# Epigenetic Regulation in Chromium-, Nickel- and Cadmium-Induced Carcinogenesis

**DOI:** 10.3390/cancers14235768

**Published:** 2022-11-23

**Authors:** Lei Zhao, Ranakul Islam, Yifang Wang, Xiujuan Zhang, Ling-Zhi Liu

**Affiliations:** Department of Medical Oncology, Sidney Kimmel Cancer Center, Thomas Jefferson University, Philadelphia, PA 19107, USA

**Keywords:** heavy metal, epigenetics, DNA methylation, histone modification, non-coding RNA, carcinogenesis, angiogenesis

## Abstract

**Simple Summary:**

Environmental and occupational exposure to heavy metals are risk factors for cancers. However, the molecular mechanisms of how heavy metals induce carcinogenesis are still not clear. In recent years, more and more studies have shown that abnormal epigenetic modifications are involved in metal-induced cancers. In this review, we summarize the up-to-date understanding of the contributions of epigenetic mechanisms to heavy metal-induced carcinogenesis and angiogenesis.

**Abstract:**

Environmental and occupational exposure to heavy metals, such as hexavalent chromium, nickel, and cadmium, are major health concerns worldwide. Some heavy metals are well-documented human carcinogens. Multiple mechanisms, including DNA damage, dysregulated gene expression, and aberrant cancer-related signaling, have been shown to contribute to metal-induced carcinogenesis. However, the molecular mechanisms accounting for heavy metal-induced carcinogenesis and angiogenesis are still not fully understood. In recent years, an increasing number of studies have indicated that in addition to genotoxicity and genetic mutations, epigenetic mechanisms play critical roles in metal-induced cancers. Epigenetics refers to the reversible modification of genomes without changing DNA sequences; epigenetic modifications generally involve DNA methylation, histone modification, chromatin remodeling, and non-coding RNAs. Epigenetic regulation is essential for maintaining normal gene expression patterns; the disruption of epigenetic modifications may lead to altered cellular function and even malignant transformation. Therefore, aberrant epigenetic modifications are widely involved in metal-induced cancer formation, development, and angiogenesis. Notably, the role of epigenetic mechanisms in heavy metal-induced carcinogenesis and angiogenesis remains largely unknown, and further studies are urgently required. In this review, we highlight the current advances in understanding the roles of epigenetic mechanisms in heavy metal-induced carcinogenesis, cancer progression, and angiogenesis.

## 1. Introduction

Environmental and occupational exposure to heavy metals has long been recognized as a global public health concern. The main routes of environmental exposure to heavy metals include contaminated food, water, and air, affecting the health of millions of people around the world [1]. However, some heavy metals are widely used in modern industries due to their special physical and chemical properties; the wide industrial use may lead to occupational exposure in workers. Heavy metal exposure may cause a broad range of toxic effects in multiple organs, such as respiratory problems, gastrointestinal (GI) dysfunction, kidney toxicity, skin lesions, and vascular damage [2]. Because heavy metals are non-biodegradable and some show persistent effects even after termination of exposure, heavy metal exposure can cause long-term health risks [3]. Among the common environmental and industrial heavy metals, hexavalent chromium (Cr(VI)), nickel (Ni), and cadmium (Cd) are classified as Group 1 carcinogens by the International Agency for Research on Cancer (IARC) [4]; exposure to them is strongly associated with increased risk of various types of cancer, such as lung cancer, breast cancer, and GI cancers [2,5]. In vitro transformation experiments and in vivo animal studies have provided solid evidence for their carcinogenic ability in transforming normal cells into cancer cells and causing cancers in animals. Multiple mechanisms have been discovered to contribute to heavy metal-induced carcinogenesis, including oxidative stress, DNA damage and repair, and abnormal signaling transduction [6]. In recent years, alterations of epigenetic modulations, such as DNA methylations, histone modifications, and non-coding RNAs, have also been shown to contribute to metal carcinogen-induced cell transformation [7]. In this review, we summarize the most up-to-date knowledge on the involvement of epigenetic mechanisms and non-coding RNAs in carcinogenic metal-induced tumorigenesis and angiogenesis.

### 1.1. Epigenetic Mechanisms

Epigenetics refers to dynamic and heritable modifications to the chromatin without alterations in the genomic sequence; epigenetic regulations represent important mechanisms in regulating gene expression. Epigenetic modifications of the genome commonly include DNA and RNA methylations, histone modifications, and non-coding RNAs. These modifications regulate gene expression by altering the local structural dynamics of chromatin, primarily regulating its accessibility and compactness [8]. Compared to genomic alterations, epigenetic alterations are usually reversible and more quickly regulated. A large number of studies have shown that cancer cells harbor extensive “epigenomic landscape” aberrations, which may lead to global dysregulation of gene expression profiles [9]. The aberrant epigenetic alterations, alone or along with widespread genetic alterations, are widely involved in cancer formation and development. In recent years, a large number of studies have shown that epigenetic modifications are involved in metal-induced carcinogenesis. In this review, we summarize the mechanisms of epigenetic modifications, including DNA methylation, histone modifications, and non-coding RNAs, in tumorigenesis and angiogenesis caused by heavy metal exposure.

#### 1.1.1. DNA Methylation and Demethylation

DNA methylation is the most extensively studied epigenetic mechanism; it generally refers to the covalent addition of methyl groups from S-adenosylmethionine (SAM) to the 5-position of the cytosine pyrimidine ring [10]. DNA methylation predominantly occurs in short CpG-rich DNA domains, called CpG islands (CGIs), which are preferentially located on the 5′ end of genes [10]. DNA methylation and demethylation are dynamic processes, which harbor a fundamental function in regulating gene expression. In general, DNA methylation leads to gene silencing because the 5-methylcytosine structure can inhibit transcriptional factors from binding to DNA and/or remodel chromatin to make it less accessible [11]. DNA methylation is catalyzed by DNA methyltransferases (DNMTs), which work cooperatively to transfer the methyl groups to the CGIs of chromatin. In contrast, DNA demethylation generally results in the re-expression of silenced genes due to DNA methylation [11] (Figure 1). Because of its fundamental role in regulating gene expression, DNA methylation/demethylation is widely involved in various cellular functions and processes, such as cell proliferation, cell cycle, apoptosis, migration, and invasion [12]. Therefore, the aberrant DNA methylation and demethylation dynamics have been shown to contribute to a variety of human diseases and cancers [12]. For example, global DNA hypomethylation may contribute significantly to the formation and development of cancers, because it predisposes cells to genomic instability and further genetic errors [13]. In addition to genome-wide hypomethylation, site-specific hyper- and hypomethylation have also been implicated in cancer development by regulating the expression of oncogenes or tumor suppressors [13]. For example, hypomethylation and increased expression of S100 calcium-binding protein A4 (S100A4) have been reported in colon cancer [14]. Silencing of RUNX3 due to promoter hypermethylation occurred in esophageal cancer, leading to the inactivation of its downstream targets [15]. In recent years, a large number of studies have shown that aberrant DNA methylation is involved in metal-induced carcinogenesis.

#### 1.1.2. Histone Modification

Here, we summarize the mechanisms of histone modifications in tumorigenesis and angiogenesis caused by heavy metal exposure. DNA is packaged into chromatin in the nucleus; chromatin is a dynamic structure composed of nucleosomes, which are the basic structural units of chromatin. Histones are the central components of the nucleosomes; four core histone proteins (H3, H4, H2A, H2B) form an octamer that is wrapped by a 147-base-pair DNA segment. Histone proteins contain a globular C-terminal domain and an unstructured extended N-terminal tail, which is densely populated with basic lysine and arginine residues [16]. This tail is usually subject to various post-translational covalent modifications (PTMs), including acetylation, methylation, ubiquitylation, phosphorylation, and SUMOylation at specific amino acidic residues [17] (Figure 2). The extensively modulated histone tail contributes to regulating gene expression by dynamically remodeling chromatin structures. Among the PTMs, acetylation and methylation of the lysine residues on H3 and H4 have been extensively studied. Acetylation of H3/H4 helps to loosen up the tight configuration of chromatin, enabling transcriptional factors access for enhanced transcription [17]. The addition and removal of acetyl groups on histones are attributed to the catalyzing activity of histone acetyltransferases (HATs) and histone deacetylases (HDACs), respectively [17]. Histone methylation is more complicated, and the effects of the methylation depend on the residues that are methylated. For example, methylation of lysine 4/36/79 in histone H3 (H3K4/36/79) is typically associated with active gene transcription, while methylation at H3K9/27 and H4K20 are generally repressive epigenetic marks [18]. The methylation and demethylation of histones are catalyzed by histone methyltransferases (HMTs) and histone demethylases (HDMTs), respectively [18]. Given the important roles of histone methylations in regulating gene expression, they are widely involved in various cellular functions and processes, such as cell proliferation, differentiation, cell cycle, and apoptosis [19]. Both global and site-specific aberrations of histone methylation are closely associated with various human diseases and cancers. For example, genome-wide loss of H4-lysine 16 acetylation (H4K16ac) has been characterized as a common hallmark of human tumors [20]. Aberrant H3K9 and H3K27 methylation patterns are associated with various forms of cancer [21]. Additionally, the timeless circadian regulator (TIMELESS) gene was transcriptionally activated by H3K27ac in colorectal cancer (CRC), and up-regulated TIMELESS promoted the proliferation, invasion, and metastasis of CRC both in vitro and in vivo [22]. In recent years, more and more studies have shown that histone modifications are involved in metal-induced carcinogenesis.

#### 1.1.3. Non-Coding RNA

The majority of the human genome is transcribed to RNAs that do not encode proteins, named non-coding RNAs. Non-coding RNAs are mainly categorized according to their size as small non-coding RNAs (< 200 nt) and long non-coding RNAs (lncRNAs, > 200 nt) [23]. Among non-coding RNAs, microRNA (miRNA), lncRNA, and circular RNA (circRNA) have been shown to play important roles in regulating gene expression at transcriptional or post-transcriptional levels; therefore, they are widely involved in the initiation and progression of human malignancies [23]. In recent years, a growing number of studies have shown that non-coding RNAs are dysregulated in cells exposed to heavy metals and play important roles in metal-induced carcinogenesis. In this review, we highlight the most recent understanding of the roles of non-coding RNAs, mainly miRNA, lncRNA, and circRNAs, in carcinogenic metal-induced cancers.

##### miRNA

A growing list of recent studies shows that miRNA dysregulations are important in metal carcinogen-induced cell malignant transformation, tumor growth, and angiogenesis. miRNAs are the most characterized non-coding RNA type; they are a large family of endogenous small non-coding RNAs with a length of 18–23 nucleotides. Generally, miRNAs promote the degradation and/or inhibit the translation of specific target messenger RNAs (mRNAs) by binding to the seed sequences in the 3′-untranslated region (3′-UTR) of the target transcript [24] (Figure 3). miRNAs have been estimated to regulate about two-thirds of all protein-coding genes in humans [24]. An individual miRNA can target many transcripts, and one mRNA molecule can be targeted by multiple miRNAs, forming a complex regulatory network. miRNAs have been shown to be involved in almost all aspects of important biological processes, such as cell proliferation and differentiation, cell death, and metabolism [25]. They are also involved in all steps in cancer development and progression [25]. Similar to protein-coding genes, miRNAs are subjected to genetic and epigenetic regulation; aberrant genetic and epigenetic mechanisms may lead to dysregulated miRNA expression, such as aberrant transcriptional regulation, DNA hypermethylation, and acetylation of promoter regions [26].

A large number of studies have shown that miRNAs play key roles in cancer development, angiogenesis, and drug resistance. miRNAs may have oncogenic or tumor-suppressive functions, depending on the property of the target genes. For example, miR-141-3p promoted prostate cancer cell proliferation by inhibiting Krüppel-like factor-9 (KLF9) expression [27]. On the contrary, miR-145 inhibited CRC growth and angiogenesis by directly targeting p70S6K1 [28]. miR-143-3p inhibited the proliferation, cell migration, and invasion of human breast cancer cells by restricting the expression of MAPK7 [29]. miRNAs are also involved in the development of drug resistance. For example, miRNA-138-5p was significantly down-regulated in lung cancer cells resistant to gefitinib (a widely used small molecule EGFR tyrosine kinase inhibitor); re-expression of miRNA-138-5p was sufficient to sensitize gefitinib-resistant lung cancer cells to gefitinib [30]. In recent years, many investigations have examined the involvement of miRNAs in metal carcinogen-caused cancers. These studies revealed that heavy metal exposure could cause global changes in miRNA expression, and some dysregulated miRNAs were involved in metal carcinogenesis.

##### lncRNA

lncRNAs are a family of non-protein coding transcripts with lengths >200 nt, which are now regarded as an important mechanism of epigenetic regulation of gene expression. lncRNAs are transcribed from intergenic, antisense, or promoter-proximal regions of the genome. lncRNAs regulate gene expression through multiple mechanisms. In the nucleus, lncRNAs mainly function through regulating chromatin remodeling, transcription, pre-mRNA splicing, and miRNA processing. In the cytoplasm, lncRNAs mainly regulate mRNA translation, protein functions, and miRNA functions (as molecular sponges) [31]. lncRNAs play important roles in multiple cellular functions and biological processes, such as cell proliferation and differentiation, apoptosis and cell cycle, cell metabolism, migration, and invasion [32]. Abnormal expression of lncRNAs is involved in all stages of cancer development, including the formation, progression, and metastasis of cancer, by exerting pro-oncogenic or tumor-suppressive roles [32]. For example, up-regulation of lncRNA LINC00152 promoted the development of non-small cell lung cancer (NSCLC) through activating EGFR, which promoted tumor cell proliferation by increasing the activity of the PI3K/Akt (AKT serine/threonine kinase 1) pathway [33]. However, some lncRNAs act as suppressors of cancer development and progression. lncRNA GAS5 has been recognized as a tumor suppressor in several types of cancers, including breast cancer, prostate cancer, lung cancer, and colorectal cancer. Down-regulated expression of GAS5 promoted proliferation and inhibited apoptosis of cancer cells [34]. In recent years, an increasing number of studies have shown that lncRNA expression is disturbed upon exposure to heavy metals and in metal-transformed cells, and dysregulated lncRNA is also involved in metal carcinogen-induced cancers. Notably, the role of lncRNAs in metal-induced cancers is understudied, and their roles in heavy metal-induced carcinogenesis need to be further investigated.

## 2. Chromium (Cr)

This section summarizes the up-to-date studies on the involvement of epigenetic mechanisms and non-coding RNAs in Cr(VI)-induced carcinogenesis and angiogenesis. Cr is a naturally occurring heavy metal widely distributed in soil, rocks, and living organisms. Cr primarily exists in two stable valence states: trivalent chromium [Cr(III)] or Cr(VI) [35]. Environmental Cr(VI) exposure is mainly through water, air, automobile exhaust, and tobacco consumption. Occupational exposure to Cr(VI) often occurs in industrial settings, such as leather tanning, smelting, welding, stainless steel production, and pigment manufacturing; the main routes of occupational Cr(VI) exposure are inhalation or dermal absorption [36]. In the blood, most Cr is bound to plasma proteins, particularly transferrin; only about 5% of Cr is free [37]. Cr accumulates mainly in the liver, spleen, soft tissue, and bone in the human body [38,39]. Cr is mainly excreted through the urinary system; therefore, urinary Cr level serves as a good indicator of Cr absorption [40,41]. Environmental and occupational Cr(VI) exposure have become major public health concerns. Cr(VI) exposure can cause a series of adverse effects on the respiratory system (including asthma, bronchitis, and respiratory tract irritation), skin (including contact dermatitis, skin burns, blisters, and skin ulcers), gastrointestinal (GI) tract (including gastric ulcers and gastritis), and kidneys (acute tubular necrosis and renal failure) [36]. Acute high-level Cr poisoning is often fatal; no proven antidote is available and the treatments are usually supportive and symptomatic, such as ventilatory support, cardiovascular support, and renal and hepatic function monitoring [42]. Orally administered ascorbic acid, chelating reagents, and exchange transfusion have been tried for acute Cr intoxication; however, the currently available evidence is not sufficient to support the application of these therapies in acute Cr poisoning [42]. For patients with chronic low-dose Cr exposure, no specific treatment is needed and the patients should avoid further exposure and rely on the urinary and fecal clearance of the accumulated Cr in the body [42]. Cr(VI) compounds are also classified as Group 1 human carcinogens by IARC [43]. As a potent human carcinogen, Cr(VI) exposure is associated with an increased risk of lung cancer among occupationally exposed workers [44,45].

The carcinogenic effects of Cr(VI) have been mainly studied in lung cancers as the lung is the major target of Cr(VI). Cr(VI) is well-known for its ability to transform normal human lung epithelial cells, such as BEAS-2B and 16HBE cells; the transformed cells show cancerous and cancer stem cell (CSC)-like properties [46]. Multiple mechanisms have been identified to contribute to Cr(VI)-induced lung carcinogenesis, including oxidative stress, DNA damage, abnormal signaling transduction, and inflammatory responses [47]. Although Cr(VI)-induced genotoxicity and mutagenicity are thought to be the primary mechanisms of Cr(VI) carcinogenesis, an increasing number of studies have shown that altered epigenetic modifications and dysregulation of non-coding RNAs contribute to Cr(VI)-induced cell transformation and tumorigenesis in recent years.

### 2.1. DNA Methylation

As a major type of epigenetic modification, DNA methylation has been intensely studied in the context of Cr(Ⅵ) exposure [48]. Fundamental alterations in DNA methylation status have been discovered in the blood and lung cancer tissues of Cr(VI)-exposed workers and in Cr(VI)-exposed and transformed lung epithelial cells [49]. Here, we summarize the most recent understanding regarding the roles of DNA methylation in Cr(VI)-induced tumorigenesis and angiogenesis.

Because DNA damage represents one of the major genotoxic effects of Cr(VI), some studies have investigated the role of DNA methylation in Cr(VI)-caused DNA damage and dysfunction of the DNA repair system. Cr(VI) exposure has been shown to cause increased DNA damage and down-regulated expression of p16INK4a (a negative regulator of the cell cycle) in 16HBE cells. The CpG1, CpG31, and CpG32 of p16INK4a were hypermethylated and the methylation levels of these sites were negatively correlated with p16INK4a expression and cell survival rate but positively correlated with DNA damage level, suggesting that increased methylation of p16INK4a may contribute to Cr(VI)-induced cancer by lowering p16INK4a expression and causing accumulation of DNA damage [50]. Furthermore, reduced expression of p16INK4a and aberrantly up-regulated methylation of p16INK4a promoter were also discovered in human workers with lung cancer and long-term (≥15 years) exposure to Cr(VI), suggesting that the hypermethylation of p16INK4a is involved in the Cr(VI) carcinogenesis [51]. In another study, the methylation was increased for the CpG sites in DNA repair genes, including O6-methylguanine-DNA-methyltransferase (MGMT), 8-oxoguanine DNA glycosylase (HOGG1), RAD51 recombinase (RAD51), X-ray repair cross-complementing 1 (XRCC1), and ERCC excision repair 3 (ERCC3). The mRNA levels of these genes were decreased in Cr(VI)-exposed 16HBE cells; the methylation levels of these genes were negatively correlated with corresponding mRNA levels [52]. In a cross-sectional study of 87 workers exposed to Cr and 30 subjects without Cr exposure, DNA damage was accumulated in Cr(VI)-exposed workers, and hypermethylation of the CpG sites of DNA repair genes, including MGMT, HOGG1, and RAD51, was also observed in these workers [52]. These studies indicate that DNA hypermethylation may suppress the DNA repair system, leading to accumulated genetic damage and finally contributing to Cr(VI) carcinogenesis.

Lung cancers of Cr(VI)-exposed workers have a higher frequency of replication error (RER) (defined by the presence of microsatellite instability (MSI)) than lung cancers of patients not exposed to Cr(VI) [53,54]. MSI is often caused by the loss of DNA mismatch repair (MMR) genes, such as MutL homolog 1 (MLH1) [55]. In the Cr(VI)-exposed lung cancers, the MLH1 level was repressed and the down-regulated MLH1 expression was correlated with the degree of MSI [54,56]. Increased methylation of MLH1 promoter was observed in Cr(VI)-exposed lung cancers, which was associated with repressed MLH1 expression [54,56,57]. Notably, the direct effect of MLH1 methylation on MLH1 protein expression was not investigated by these studies. The overall methylation status was also significantly higher in Cr(VI) lung cancers than non-Cr(VI) lung cancers [57]. Additionally, increased promoter methylation and decreased expression of p16 and APC were also observed in Cr(VI)-exposed lung cancers [57]. These results suggest that DNA hypermethylation-mediated down-regulation of MLH1 expression may contribute to Cr(VI)-induced carcinogenesis by impairing the DNA repair system. The roles of aberrant DNA methylation/demethylation in Cr(VI)-induced carcinogenesis and cancer progression are summarized in Table 1.

### 2.2. Histone Modification

#### 2.2.1. Histone Methylation

Histone modification is another common type of epigenetic modification involved in the formation and development of cancers [16]. Cr(VI) exposure has been shown to cause extensive alterations in histone modifications. For example, Cr(VI) exposure caused globally increased levels of H3K9me2/3 and H3K4me2/3 but decreased levels of H3K27me3 and H3R2me2 in A549 human lung cancer cells [58,59]. Cr(VI) also caused gene-specific histone modifications that resulted in altered gene expression. For example, Cr(VI) exposure induced increased H3K9me2 in the promoter of MLH1, likely through up-regulating G9a (also known as euchromatic histone-lysine N-methyltransferase, 2), a histone methyltransferase that specifically methylates H3K9, causing repressed MLH1 expression [58]. The altered global and gene-specific histone modifications and the resultant gene expression changes, such as inhibition of tumor suppressor MLH1, have been shown to contribute to Cr(VI) carcinogenesis in a growing number of studies.

Chronic low-dose Cr(VI) exposure has been shown to induce cell transformation and acquisition of cancer stem cell (CSC)-like properties in BEAS-2B and 16HBE cells. Cr(VI)-transformed cells had increased levels of H3K9me2 and H3K27me3, two repressive methylation marks, and greatly increased levels of histone-lysing methyltransferases (HMTs) for H3K9 (GLP, G9a, SUV39H1 histone lysine methyltransferase (SUV39H1)) or H3K27 (enhancer of zeste 2 polycomb repressive complex 2 subunit (EZH2)) [46]. Additionally, SUV39H1 and EZH2 were also up-regulated in the lung cancer of Cr(VI)-exposed workers [46]. The up-regulated HMTs played a causal role in elevated levels of H3K9me2 and H3K27me3, and were indispensable in Cr(VI)-induced cell transformation and in maintaining the cancerous and CSC-like property of Cr(VI)-transformed cells. Furthermore, increased HMTs also contributed to Cr(VI)-exposure-caused DNA damage, as knockdown of HMTs attenuated Cr(VI)-induced DNA damage. Collectively, these results indicate that dysregulated histone modification machinery contributes to Cr(VI)-induced genotoxic effects, carcinogenesis, and cancer progression [46].

Hedgehog (Hh) signaling plays a key role in embryogenesis and stem cell functions; dysregulation of the Hh pathway has been reported in various human cancers. The expression of the hedgehog-interacting protein (HHIP), a downstream target and a negative regulator of Hh signaling, was down-regulated in Cr(VI)-transformed BEAS-2B cells and primary lung cancers [60]. The down-regulation of HHIP contributed to Cr(VI)-induced malignant transformation by activating the Hh signaling, as the forced expression of HHIP inactivated Hh signaling and inhibited cell proliferation and anchorage-independent growth in Cr(VI)-transformed cells [60]. Mechanistically, the suppressed HHIP expression was attributed to multiple epigenetic modifications of its promoter region, including DNA hypermethylation, reduced levels of H3K9ac and H3K4me3, two active histone marks, and enriched H3K27me3, a repressive histone mark [60].

#### 2.2.2. Histone Acetylation

Histone acetylation is another major type of histone modification. C-X-C motif chemokine ligand 5 (CXCL5) is an inflammatory factor involved in multiple processes relevant to tumor formation and progression [61]. CXCL5 level was dramatically up-regulated in the peripheral blood monocytes (PBMCs) and plasma from workers with occupational exposure to Cr(VI) and the lung tissues of mice intranasally exposed to Cr(VI) and Cr(VI)-transformed BEAS-2B cells [62]. Functionally, CXCL5 promoted Cr(VI)-induced cell transformation and played an important role in maintaining cancer phenotypes of Cr(VI)-transformed BEAS-2B cells. CXCL5 also prompted Cr(VI)-induced epithelial-mesenchymal transition (EMT) by up-regulating zinc finger E-box binding homeobox 1 (ZEB1) [62]. Two mechanisms have been proposed to account for the up-regulated CXCL5: Cr(VI) exposure activated c-Myc, which specifically binds to the CXCL5 promoter and recruited p300 to form a transcription complex; the c-Myc/p300 complex then enhanced the histone H3 acetylation and eventually promoted CXCL5 transcription. Hypomethylation of CXCL5 promoter due to Cr(VI)-induced DNA methyltransferase 1 (DNMT1) down-regulation also contributed to CXCL5 induction [62]. These findings indicate that epigenetic machinery contributes to Cr(VI)-induced carcinogenesis and cancer progression through epigenetically up-regulating CXCL5. The level of nuclear protein 1 (NUPR1) was significantly up-regulated in Cr(VI)-exposed BEAS-2B cells due to epigenetic mechanisms, such as hypomethylation and increased H3K9 and H3K14 acetylation levels of its promoter [63]. Elevated NUPR1, in turn, led to reduced global and promoter-specific H4K16ac (such as for tripartite motif containing 42 (TRIM42) and inhibitors of apoptosis (IAP)) by inhibiting the transcription of males absent on the first (MOF), a histone acetyltransferase that specifically acetylates H4K16 [64]. Functionally, up-regulated NUPR1 promoted Cr(VI)-induced transformation of BEAS-2B cells, consistent with previous reports that NUPR1 promoted the development and metastasis of lung, pancreatic, and breast cancers [65,66,67]. Notably, NUPR1 expression was not increased in transformed BEAS-2B cells, suggesting that it may only be required for initiation of cell transformation; however, reduced MOF and H4K16ac were observed in both Cr(VI)-exposed and transformed cells, indicating that they were required for both initiation and maintenance of cell transformation [63]. These results support the notion that Cr(VI)-induced NUPR1 contributes to Cr(VI)-induced carcinogenesis by altering the histone modification marks and subsequent gene expression perturbations [63]. c-Myc is a master regulator of cell metabolism; dysregulated metabolism plays an important role in cancer development [68]. Clementino et al. have found that c-Myc mediated aberrant histone acetylation contributes to Cr(VI)-induced carcinogenesis by promoting glycolytic shift [69]. Cr(VI)-transformed BEAS-2B cells showed glycolytic shift mediated by up-regulated c-Myc. The glycolytic shift, in turn, led to increased acetyl coenzyme A (acetyl-CoA) levels and several histone acetylation marks, including H3K9ac, H3K27ac, and acetyl-histone 4 (Ac-H4) and 2B (Ac-H2B). The up-regulated acetylation of H3 at the promoter of ATP citrate lyase (ACLY) promoted the transcription of ACLY, a key enzyme for producing acetyl-CoA; up-regulated ACLY, in turn, increased c-Myc expression, acetyl-CoA level, and histone acetylation, forming a positive feedback loop that drives a metabolic shift in Cr(VI)-transformed cells [69]. The glycolytic shift and resultant glycolysis played critical roles in maintaining the malignant phenotypes of Cr(VI)-transformed cells, because the reverse of the glycolytic shift by glucose depletion significantly inhibited the growth, CSC-like property, and tumorigenicity of the transformed cells [69]. SET nuclear proto-oncogene (SET), a major regulator of histone modifications, was increased in Cr(VI)-exposed and Cr(VI)-transformed 16HBE cells [70]. Increased SET promoted proliferation and cell cycle progression and inhibited apoptosis of transformed 16HBE cells. Mechanistic studies indicated that increased SET mediated the reduction in H3K18ac and H3K27ac at the tumor protein P53 binding protein 1 (53BP1) promoter, resulting in decreased expression of 53BP1 in Cr(VI)-exposed and Cr(VI)-transformed 16HBE cells [70,71]. The 53BP1 protein binds to the central domain of TP53 and plays an important role in DNA damage repair [72]; inhibition of 53BP1 may cause DNA damage accumulation and inhibition of apoptosis, therefore promoting tumorigenesis and cancer progression [73]. Additionally, Cr(VI) also caused global decreases in H3K18ac and H3K27ac, which might have broad effects on gene expression. These results demonstrate the involvement of SET-mediated histone hypoacetylation in Cr(VI)-induced lung carcinogenesis. The roles of histone modifications in Cr(VI)-induced carcinogenesis and cancer progression are summarized in Table 2.

### 2.3. Non-Coding RNA

#### 2.3.1. miRNA

The dysregulation of miRNA expression has recently been shown to have important roles in Cr(VI)-induced cell transformation, carcinogenesis, and angiogenesis. Speer et al. reported that acute or prolonged exposure to Cr(VI) led to altered global miRNA expression in the human bronchial fibroblast WTHBF-6 cell line. In silico pathway analysis revealed these altered miRNAs were enriched in pathways involved in carcinogenesis [74]. Two redox-sensitive miRNAs, miR-27a and miR-27b, were down-regulated in response to ROS production in BEAS-2B cells chronically exposed to Cr(VI), in Cr(VI)-transformed BEAS-2B cells, and in the lung tissues of mouse intranasally exposed to Cr(VI) for 12 weeks. The down-regulated miR27a/b promoted Cr(VI)-induced tumorigenesis and angiogenesis through up-regulating NF-E2-related factor-2 (Nrf2) [75], a transcription factor that has been known to promote cell proliferation, colony formation, migration, and angiogenesis in various cancers [76]. These results indicate that miR-27a and miR-27b act as tumor suppressors and the miR-27a/b/Nrf2 signaling plays a pivotal role in Cr(VI)-induced carcinogenesis and angiogenesis.

An oncogenic miRNA, miR-21, was up-regulated in Cr(VI)-treated BEAS-2B cells, and elevated miR-21 promoted Cr(VI)-induced transformation of BEAS-2B cells. Mechanistically, Cr(VI) induced interleukin-6 (IL-6) expression, which in turn promoted the signal transducer and activator of transcription 3 (STAT3) phosphorylation, and activated STAT3 bound to the miR-21 promoter to promote its transcription. Up-regulated miR-21 directly down-regulated programmed cell death 4 (PDCD4), a tumor suppressor [77,78]; inhibition of PDCD4 suppressed downstream E-cadherin expression and promoted β-catenin/TCF-dependent transcription of c-Myc and plasminogen activator, urokinase receptor (uPAR). The c-Myc oncogene plays a critical role in the process of carcinogenesis [79], and uPAR is involved in cancer progression [80]. These results suggest that activation of the miR-21/PDCD4 signaling contributes to Cr(VI)-induced lung carcinogenesis through regulating important downstream factors, such as E-cadherin, c-Myc, and uPAR [77]. Additionally, up-regulated miR-21 and suppressed PDCD4 were also observed in the lung tissues of mice intranasally exposed to Cr(VI), multiple lung cancer cell lines (H2030, H460, H23, and A549), and human lung adenocarcinoma tissues, suggesting that the miR-21/PDCD4 pathway is involved in lung cancer caused by both Cr(VI) and non-Cr(VI) factors [77]. Furthermore, quercetin, an antioxidant flavonoid widely present in fruits and vegetables, inhibited Cr(VI)-induced activation of miR-21/PDCD4 cascade in BEAS-2B cells by decreasing ROS generation; quercetin therefore inhibited Cr(VI)-induced malignant transformation and suppressed the growth of xenograft tumor of Cr(VI)-transformed cells, suggesting a preventive and therapeutic role of quercetin in Cr(VI)-caused lung cancer [78]. On the contrary, Cr(VI) caused the down-regulation of miR-21 in L02 hepatocytes, and the subsequent increase in PDCD4 contributed to Cr(VI)-induced apoptosis and inhibited proliferation of L02 cells, suggesting that the inhibited miR-21/PDCD4 signaling contributes to Cr(VI)-induced hepatotoxicity [81].

Expression of miR-143, a tumor-suppressive miRNA [82], was decreased in BEAS-2B cells exposed to Cr(VI), Cr(VI)-transformed BEAS-2B cells, and the plasma of Cr(VI)-exposed workers [83,84]. Suppressed miR-143 promoted the growth and tumor angiogenesis of Cr(VI)-transformed BEAS-2B cells in vitro and in vivo [83,84]. Mechanistically, ectopic miR-143 overexpression directly targeted both insulin-like growth factor-1 receptor (IGF-IR) and insulin receptor substrate-1 (IRS1), suppressed the activation of downstream ERK signaling, and inhibited the EKR pathway, lowering the expression of IL-8, a major angiogenesis activator in Cr(VI)-induced angiogenesis [83]. miR-143 could also inhibit the expression of IL-6, hypoxia-inducible factor-1 subunit alpha (HIF-1α), p70S6K1, and NF-κB p65, causing reduced expression of IL-8 and vascular endothelial growth factor (VEGF) [83,84]. Collectively, these results indicate that down-regulation of miR-143 contributes to Cr(VI)-induced carcinogenesis and angiogenesis through activation of multiple signaling pathways, including IGF-IR/IRS1/ERK, mTOR/p70S6K1, HIF-1α/VEGF, and NF-κB p65 pathways [83,84]. Moreover, the miR-143 level was also significantly lower in A549 and H2195 lung cancer cells than in BEAS-2B cells, suggesting a broad role of miR-143 in lung cancer [83].

Cr(VI) exposure has been shown to cause DNA damage and subsequent activation of DNA repair genes. In a cohort study, plasma miR-3940-5p level was significantly down-regulated in workers exposed to Cr(VI) and negatively associated with blood Cr level. However, the low miR-3940-5p level was associated with high expression of XRCC2 (a DNA repair gene targeted by miR-3940-5p) in peripheral lymphocytes [85]. Furthermore, miR-3940-5p was inhibited in Cr(VI)-treated 16HBE cells; down-regulated miR-3940-5p then enhanced homologous recombination after double-strand breaks (DSBs) caused by Cr(VI) exposure [86]. These studies suggest that repression of miR-3940-5p plays a protective role in Cr(VI)-induced cell transformation by mitigating the accumulation of DNA damage. Exposure of human B lymphoblast HMy2.CIR cells to Cr(VI) caused global miRNA expression changes. Functional analysis of altered miRNAs indicated that down-regulation of miR-148a-3p and miR-21-5p might contribute to Cr (VI)-induced cell apoptosis, and up-regulated miR-221-3p might result in increased cell apoptosis and accumulation of DSBs and total DNA damage [87]. c-Myc, a proto-oncogene, played a critical role in maintaining the CSC-like property and tumorigenicity of Cr(VI)-transformed BEAS-2B cells; it also promoted Cr(VI)-induced transformation and acquisition of CSC-like properties in BEAS-2B cells. Mechanistic studies indicated that chronic Cr(VI) exposure increased c-Myc expression by down-regulating the level of miR-494, suggesting that inhibited miR-494/c-Myc cascade contributes to chronic Cr(VI) exposure-induced initiation and progression of lung cancer [88]. The roles of miRNAs in Cr(VI)-induced carcinogenesis and cancer progression are summarized in Table 3.

#### 2.3.2. lncRNA

Fewer studies have been performed to investigate the role of lncRNAs in Cr(VI)-induced cancers. Hu et al. characterized the differentially expressed lncRNAs in 16HBE cells exposed to Cr(VI) for 24 h. They found 1868 significantly up-regulated and 2203 significantly down-regulated lncRNAs. Further bioinformatics analysis suggested that the differentially expressed lncRNAs formed a complex regulation network and were associated with immune response, cell cycle, DNA damage, repair, etc. However, whether these dysregulated lncRNAs contributed to Cr(VI)-induced carcinogenesis is currently unknown [89].

## 3. Nickel (Ni)

Ni is a ferromagnetic heavy metal widely distributed in the environment, such as soil, air, and water. Ni has a variety of toxic effects upon environmental or occupational exposure. Environmental exposure is mainly through contaminated soil, food, water, and air. Due to its unique physical and chemical properties, Ni is widely used in modern metallurgical industries, such as alloy production, electroplating, and battery production. Occupational exposure generally occurs in these industrial settings, threatening the health of millions of workers worldwide [90]. Ni can enter the human body through multiple routes, including inhalation, ingestion with food or water, and dermal absorption, depending on the chemical form of the pollutant [91]. Inhalation is the most common and riskiest route of exposure to Ni, which is associated with an increased risk of lung cancer [92]. The inhaled Ni particles are deposited in different locations of the respiratory system, depending on the diameter, solubility, and quantity of the particles. Water-soluble Ni compounds are absorbed by the lungs and enter the blood and finally are excreted by the kidneys. Insoluble Ni compounds stay in the respiratory system for a much longer time, where they cause respiratory manifestations and even lung cancer [90]. Exposure to Ni can cause a variety of toxic effects on human health, such as contact dermatitis, allergy, asthma, cardiovascular dysfunction, kidney diseases, lung fibrosis, and lung and nasal cancer, depending on the dose, length, and chemical properties of exposed chemicals [93,94]. Topical corticosteroids and nonsteroidal creams can be applied to reduce irritation and rashes; for severe allergic scenarios, oral corticosteroids and antihistamines can be used [95]. Acute Ni poisoning can be treated with chelating reagents, the same as the treatment of the poisoning of other heavy metals. Sodium diethyldithiocarbamate (Dithiocarb) has been proved to be most effective, while other chelators, such as tetraethylthiuram, d-penicillamine, and dimercaprol, are less effective and therefore have limited therapeutic value for Ni poisoning [96]. Ni compounds have also been classified as Group 1 carcinogens by IARC [4,6]. Long-term exposure to Ni is known to cause multiple types of human cancers, including lung and nasal cancers [97]. In the past decades, researchers have put a lot of effort into revealing the molecular mechanisms of Ni carcinogenesis. A large number of studies have shown that Ni has low mutagenic potential, and Ni acts predominantly through epigenetic mechanisms including DNA methylation, histone modification, and non-coding RNAs to cause cancers. This section reviews the up-to-date understanding of epigenetic mechanisms in Ni-induced cancers.

### 3.1. Ni and DNA Methylation

An early study using the transgenic gpt+ G12 Chinese hamster cell line as a model discovered that Ni could alter gene expression by enhancing DNA methylation and chromatin condensation, rather than by mutagenic mechanisms [98,99]. In recent years, more and more studies focus on epigenetic modifications in Ni-induced carcinogenesis. Silencing of O-6-methylguanine-DNA MGMT was observed in Ni-transformed 16HBE cells. Epigenetic modifications at the MGMT promoter, including DNA hypermethylation, reduced histone H4 acetylation and H3K9ac (indicators of open chromatin and active transcription) and up-regulated H3K9me2 (a marker of condensed and inactive chromatin), accounted for the suppression of MGMT expression in Ni-transformed cells [100]. Additionally, DNMT1, together with methyl-CpG-binding protein 2 (MECP2) and methylated DNA-binding domain protein 2 (PRDM2), were recruited to the CpGs region of the MGMT promoter to maintain the hypermethylation state [100]. These results suggest that impaired DNA repair caused by multiple epigenetic modifications may contribute to Ni-induced malignant transformation.

The acquisition of EMT has been shown to play an important role in diseases associated with Ni exposure, such as asthma, fibrosis, and lung cancer [101]. Similar to Arsenic (As) and Cr(VI), Ni has also been shown to induce EMT in BEAS-2B cells by altering EMT markers such as the up-regulation of fibronectin and down-regulation of E-cadherin [102]. Mechanistically, Ni inhibited E-cadherin expression by induction of ROS-dependent promoter hypermethylation of E-cadherin. These results suggest the involvement of epigenetic induction of EMT in Ni carcinogenesis [102].

In a Ni-induced muscle tumor model in Wistar rats, the mRNA expressions of retinoic acid receptor beta (RAR-β2), Ras association domain family member 1 (RASSF1A), and cyclin-dependent kinase inhibitor 2A (CDKN2A) were decreased, and hypermethylation of the 5’ region of these genes was found in the muscle tumors [103]. In a Ni-caused malignant fibrous histiocytomas moue model, tumors had a down-regulated expression of p16Ink4a and hypermethylation of its DNA, and activation of the MAPK signaling pathway was observed, suggesting a role of synergistic interactions between MAPK activation and p16Ink4a silence in Ni carcinogenesis [104]. Notably, neither the significance of the aberrantly expressed genes nor the influence of hypermethylation on the expression of these genes was investigated in these two studies. Bypassing senescence is a critical step in the malignant transformation of mammalian cells. A study in primary dermal fibroblast SHD cells indicated that epigenetic silencing of p16Ink4a by promoter DNA methylation contributed to Ni-induced immortalization, suggesting epigenetic silencing of p16Ink4a plays an important role in Ni carcinogenesis [105].

Activation of oncogenic genes through DNA hypomethylation is also involved in Ni-elicited cancers. Angiopoietin-like protein 4 (ANGPTL4) is induced by hypoxia, and its high expression is associated with progression and poor prognosis in several types of cancers [106,107]. Kang et al. found that ANGPTL4 was up-regulated in Ni-treated BEAS-2B and lung cancer cell lines [108]. Mechanistic studies indicated that Ni induced accumulation of HIF-1α, which mediated the transcriptional activation of ten-eleven translocation methylcytosine dioxygenase 1 (TET1), a demethylase that mediates the first step toward DNA demethylation by converting 5-mC into 5-hmC [109,110]. The up-regulated TET1 caused hypomethylation of the ANGPTL4 promoter, resulting in up-regulated ANGPTL4 expression. Because of the oncogenic property, up-regulated ANGPTL4 was supposed to contribute to Ni-elicited carcinogenesis [110]. The roles of aberrant DNA methylation/demethylation in Ni-induced carcinogenesis and cancer progression are summarized in Table 4.

### 3.2. Histone Modification

In recent years, a growing number of studies have indicated that multiple forms of histone modifications are involved in Ni carcinogenesis by regulating the expression of cancer genes and the activity of critical signaling pathways in cancer formation and development.

#### 3.2.1. Histone Methylation

Histone methylation is an important epigenetic mechanism in regulating gene transcription. Ni induced extensive long-term transcriptional changes in BEAS-2B cells even after the termination of exposure; meanwhile, genome-wide increases in H3K4me3 (an activating histone modification) and H3K27me3 (a repressive histone modification) were observed, suggesting that histone modifications are associated with Ni-induced extensive gene expression [111]. H3K9me2 is a mark for gene silencing and is organized into large repressive domains in close proximity to active genes [16,112]. Jose et al. showed that Ni caused disrupted H3K9me2 domains and subsequent spreading of H3K9me2 into active chromatin regions likely through loss of CTCF-mediated insulation. The spreading of H3K9me2 was associated with gene silencing. These results showed the effect of Ni on H3K9me2 dynamics and the resultant chromatin domain disruption [112]. Exposure of A549 cells to Ni led to significantly increased global levels of H3K9me2 [59,113] and H3K4me3 [59,114]. Specifically, increased H3K4me3 was discovered in both the promoter and coding regions of carbonic anhydrase 9 (CA9) and N-Myc downstream regulated 1 (NDRG1), which were the highest up-regulated genes in Ni-exposed A549 cells [114]. H3K27me3 is involved in Ni-induced EMT. Jose et al. indicated that Ni exposure induced permanent EMT in BEAS-2B cells through irreversible activation of ZEB1, a master EMT driver. Ni exposure led to a significantly decreased H3K27me3 level at the ZEB1 promoter compared with untreated BEAS-2B cells. Because H3K27me3 is associated with transcriptional silencing of nearby genes, loss of H3K27me3 likely caused increased ZEB1 expression, finally resulting in EMT in Ni-exposed cells [115]. In another study, Ni exposure caused increased H3K9me2 levels at the sprouty RTK signaling antagonist 2 (SPRY2) promoter by inhibiting histone demethylase Jumonji domain containing 1A (JMJD1A), resulting in repressed SPRY2 expression in BEAS-2B cells. SPRY2 is a negative regulator of ERK signaling; repression of SPRY2 thus potentiated the Ni-induced ERK phosphorylation to promote anchorage-independent growth. These results suggest that decreased expression of SPRY2 by suppressive histone methylation may contribute to Ni carcinogenesis [116].

Iron- and 2-oxoglutarate-dependent dioxygenases are a broad family of non-heme iron-containing enzymes that catalyze various important oxidative reactions in cells [117]. A key structural motif of these enzymes is a facial triad of 2-histidines-1-carboxylate that coordinates the Fe(II) at the catalytic site [117]. JMJD1A is a member of this family of dioxygenase and specifically demethylates H3K9me1/2 [118,119]. JMJD1A is a hypoxic response gene that is induced by HIF-1α and involved in repressing the transcription of target genes [118,119]. Abnormal expression of JMJD1A has been shown to contribute to cancer formation and development [120]. Ni has been shown to inhibit JMJD1A and DNA repair enzyme ABH2 (both belong to iron-dependent dioxygenase) by replacing the ferrous iron in the catalytic centers. Given the functions of these two dioxygenases, inhibition of them by Ni may cause widespread impacts on cells, such as impaired epigenetic profile and DNA repair [121]. Ni induced histone demethylase JMJD1A expression in 786-0 renal cancer cells and HEK293 cells; ascorbate could antagonize Ni-induced JMJD1A expression in these cells by decreasing the stability of HIF-1α protein [122]. Guo et al. found that short-term Ni exposure in HEK293T and 786-0 caused a reduced level of H3K27me3 and up-regulated the expression of Jumanji domain-containing protein 3 (JMJD3), an H3K27me3 demethylase, suggesting that increased JMJD3 accounts for the un-regulated H3K27me3 [123]. Chen et al. showed that Ni exposure increased global H3K9me2 by inhibiting the demethylation process, likely via decreasing the activity of a Fe(II)-2-oxoglutarate-dependent histone H3K9 demethylase [124].

The LSD1 histone demethylase complex has been shown to repress gene transcription by catalyzing the demethylation of H3K4me2, a mark for active transcription; Scm-like with four mbt domains 1 (SFMBT1) is a “histone reader” subunit of the LSD1 complex [125]. Tang et al. found that exposure of A549 cells to Ni induced EMT; furthermore, the down-regulated E-cadherin was caused by SFMBT1-mediated recruitment of LSD1 to E-cadherin promoter and subsequent decrease in H3K4me2 levels of E-cadherin promoter, suggesting the involvement of SFMBT1-LSD1-histone demethylation in the Ni-induced EMT [125]. Nicotinamide N-methyltransferase (NNMT) has been shown to decrease histone methylation by decreasing the cellular S-adenosylmethionine (SAM)/S-adenosylhomocysteine (SAH) ratio [126]. Ni has been shown to induce H3K9me2 by inhibiting NNMT expression to increase the cellular SAM/SAH ratio [127].

#### 3.2.2. Histone Acetylation

Acetylation is another important type of modification to histone. Decreased histone acetylation is associated with suppressed gene expression; therefore, it plays an important role in tumorigenesis and development. Ni has been shown to cause reduced acetylation of H2A, H2B, H3, and H4 in various cell lines, including A549, HAE, NRK, Hep3B, and IGROV1 cells [113,128,129,130], and loss of histone acetylation was associated with marked suppression of global gene transcription [128]. Further studies indicated that Ni inhibited the acetylation of histone H4 through binding to the anchoring site on histidine-18 within the NH2-terminal tail of H4 [129,131], causing conformational changes that implicated in regulating gene transcription. Other mechanisms also contribute to Ni-induced chromatin hypoacetylation. Ni induced histone hypoacetylation in Hep3B cells by inhibiting the overall histone acetyltransferase (HAT) activity, a process dependent on Ni-induced ROS production [132]. Histone hypoacetylation has been shown to play important roles in regulating cell cycle progression and apoptosis, two hallmarks of cancer. Ni exposure caused cell growth inhibition and apoptosis and significantly reduced Bcl-2 expression in Hep3B cells. A global decrease in histone acetylation and reduced histone H4 acetylation in the Bcl-2 promoter region was also observed. The decreased histone acetylation status contributed to Ni-caused apoptosis and Bcl-2 down-regulation, indicating that histone hypomethylation may be involved in Ni carcinogenesis through regulating critical genes and cellular processes in cancer [133]. Zhang et al. investigated the effect of enhanced histone acetylation on Ni-induced cell transformation by treating different cells with histone deacetylase inhibitor trichostatin A (TSA). TSA inhibited the ability of Ni to transform both human osteoblastic TE 85 cells and mouse embryo fibroblast PW cells. TSA was also able to reverse the cancer phenotypes of Ni-transformed cells. However, TSA had little or no effect on established A549 and H460 cancer cell lines in terms of forming colonies in soft agar, suggesting that the reversal effect of TSA is specific to Ni-transformed cells. These results also indicate that augmented histone acetylation has an inhibitory effect on Ni carcinogenesis and suggest the role of histone deacetylase inhibitors in preventing and treating Ni-caused cancers [134].

Ni has also been shown to regulate histone phosphorylation. Ke et al. found that Ni could induce phosphorylation of histone H3 at its serine 10 residue (H3S10) in A549 and BEAS-2B cells through activating the c-Jun N-terminal kinase (JNK)/stress-activated protein kinase (SAPK) signaling pathway [135]. H3S10 phosphorylation has been shown to facilitate cellular malignant transformation and participate in fundamental cellular functions in various human cancers [136]. For example, STAT3 activated the NFAT signaling by promoting histone H3S10 phosphorylation in the promoter of the nuclear factor of activated T cells 2 (NFATC2) in gastric carcinogenesis [137]. Therefore, alterations in H3S10 phosphorylation might contribute to the carcinogenesis caused by Ni, a notion that needs to be examined in future studies.

Ni induced substantial increases in the ubiquitination of H2A and H2B (uH2A and uH2B) in various cell lines likely through inhibiting the activities of histone deubiquitinating enzymes [113,138,139]. Ni has also been shown to cleave histone H2A [140,141] and H2B [142]. However, the influence of Ni-induced histone ubiquitination and cleavage on gene expression and carcinogenesis needs to be investigated further. The roles of histone modifications in Ni-induced carcinogenesis and cancer progression are summarized in Table 5.

### 3.3. Non-Coding RNA

#### 3.3.1. miRNA

In recent years, more and more miRNAs have been identified to play a caustic role in Ni-induced cancers. For example, Wu et al. have shown that miR-4417 plays an oncogenic role in Ni-induced lung cancer. They found that miR-4417 was significantly up-regulated in BEAS-2B and A549 cells exposed to Ni, and induction of miR-4417 contributed to the Ni-induced tumorigenesis through targeting TGF-beta activated kinase 1 (MAP3K7) binding protein 2 (TAB2) [143]. In a study by Zhu et al., the expression level of miR-31 was down-regulated in Ni-transformed BEAS-2B cells due to transcriptional repression by RUNX family transcription factor-2 (RUNX2). Inhibition of miR-31 resulted in the acquisition of cancer hallmarks in BEAS-2B cells, such as increased colony-forming ability in soft agar assay and enhanced migration and invasion abilities, whereas overexpression of miR-31 repressed these cancer hallmarks in Ni-transformed cells [144]. Further studies indicated that down-regulation of miR-31 resulted in increased expression of SATB homeobox 2 (SATB2) [144], which is a direct target of miR-31 and an important mediator of Ni-induced cell transformation [145]. These results indicate that activation of the RUNX2/miR-31/SATB2 cascade contributes to Ni-mediated lung cancers. In a human cohort study including 76 never-smoking lung cancer patients, higher miR-21 levels were associated with Ni exposure. Patients with high nickel/miR-21 levels had significantly shorter overall survival (OS) and relapse-free survival (RFS) periods compared with those with low Ni/miR-21 levels [146]. Mechanistically, Ni exposure induced miR-21 expression via activating the EGFR/NF-κB signaling pathway; up-regulated miR-21, in turn, suppressed the expression of two miR-21 target genes, SPRY2 and reversion-inducing cysteine-rich protein with kazal motifs (RECK), and promoted invasiveness of lung cancer cells [146]. Zhang et al. established a Ni-induced muscle tumor model by injecting Ni_3_S_2_ compounds into Wistar rats. The expression of miR-222 was significantly up-regulated in the muscle tumor tissues, while the expression levels of two target genes of miR-222, cyclin-dependent kinase inhibitor 1B (CDKN1B) and CDKN1C, were down-regulated in the tumors [147]. They also found that in Ni-transformed 16HBE cells, miR-222 was up-regulated with a concomitant decrease in the expression of CDKN1B and CDKN1C. Given the tumor-suppressive roles of CDKN1B and CDKN1C, miR-222-mediated down-regulation of CDKN1B and CDKN1C may contribute to Ni-induced tumorigenesis or tumor progression [147].

The cross-talk between DNA methylation and miRNA expression has been shown to contribute to Ni-induced carcinogenesis. Zhang et al. have shown that the down-regulation of miR-203 in Ni-transformed 16HBE cells is likely due to hypermethylation of the CpGs in the miR-203 promoter and first exon region [148]. Additionally, miR-203 has been shown to suppress Ni tumorigenesis by negatively regulating its target gene ABL proto-oncogene 1 (ABL1), suggesting that DNA methylation-mediated silencing of tumor-suppressive miRNAs contributes to Ni-induced cancer [148]. Ji et al. have shown that in Ni-transformed 16HBE cells, the expression of miR-152 was down-regulated due to promoter DNA hypermethylation; inhibition of miR-152 expression significantly promoted cell growth in both parental and transformed 16HBE cells. Moreover, inhibition of miR-152 in 16HBE cells resulted in increased expression of DNMT1, which increased the methylation of the miR-152 promoter, demonstrating a feedback loop between miR-152 and the DNMT1 to further promote Ni-induced cell malignant transformation [149].

Dysregulated miRNAs are also implicated in Ni-caused metabolism shifts. He et al. have shown that Ni exposure caused a significant accumulation of HIF-1α in Neuro-2a, a mouse neuroblast cell line. Up-regulated HIF-1α, in turn, promoted the transcription of miR-210, and the elevated miR-210 modulated the energy metabolism shift to aerobic glycolysis after Ni exposure through inhibiting iron–sulfur cluster assembly proteins (ISCU1/2) [150]. ISCU1/2 mediates the assembly of ISCs, the prosthetic groups critical for enzymes that are responsible for mitochondrial respiration and energy production [151,152]. Therefore, the suppression of ISCU1/2 by miR-210 led to the inactivation of ISC-containing enzymes and ultimately inhibited mitochondrial respiration [150]. Given that enhanced aerobic glycolysis is a hallmark of cancer [153], the miR-210-mediated energy metabolism shift upon Ni exposure may contribute to Ni-induced carcinogenesis. Notably, this HIF-1α/miR-210/ISCU1/2/energy metabolism regulatory axis has been reported in different cell types [154,155], including BEAS-2B cells [156]. Additionally, melatonin could attenuate the Ni-induced increment in aerobic glycolysis in BEAS-2B cells through ROS scavenging and subsequent suppression of this regulatory axis [156]. Saquib et al. have shown that in HepG2 cells exposed to Ni nanoparticles, the expression of miR-210 was up-regulated, but the function of elevated miR-210 was not investigated [157]. The roles of altered miRNAs in Ni-induced cancers are summarized in Table 6.

#### 3.3.2. LncRNA

Reduced expression of lncRNA maternally expressed gene 3 (MEG3), a tumor-suppressive lncRNA, has been reported in multiple cancers [158]. In BEAS-2B cells exposed to Ni, MEG3 was also significantly down-regulated, which has been shown to contribute to the Ni-induced malignant transformation of BEAS-2B cells [158]. Mechanistically, Ni-induced down-regulation of MEG3 was attributed to promoter hypermethylation mediated by Ni-mediated DNMT3b overexpression, and suppression of MEG3 consequently decreased interaction of MEG3 with c-Jun, an inhibitory transcription factor for PH domain and leucine-rich repeat protein phosphatase 1 (PHLPP1), causing c-Jun-mediated PHLPP1 transcriptional inhibition. PHLPP1 suppression, in turn, activated the Akt/p70S6K/S6 axis to up-regulate HIF-1α protein translation, resulting in the malignant transformation of BEAS-2B cells [158]. These results highlight the role of the cross-talk between epigenetic modifications and non-coding RNAs in Ni-induced lung tumorigenesis.

## 4. Cadmium (Cd)

Cd and its compounds ubiquitously exist in the environment, food, drinking water, and contaminated occupational workplaces, such as factories for producing batteries and color pigments [159]. Tobacco smoking is also an important source of Cd exposure [160]. In the blood, the majority of Cd is bound to carrier proteins, such as albumin and metallothionein. Cd can be taken up by the liver, where Cd induces the production of metallothionein and causes hepatocyte necrosis and apoptosis. The main organ for long-term Cd accumulation is the kidney [161]. Urinary excretion represents the major mechanism of Cd elimination from the body, although it is very slow. Due to slow excretion, Cd can accumulate in animals and plants with an exceptionally long half-life (about 25–30 years), causing Cd contamination to continue to be a major public health concern for decades even when the exposure is terminated. Cd exposure may result in damage and dysfunction in multiple organ systems, such as the skeleton, kidney, digestive, and respiratory systems. For example, Cd has been shown to cause skeletal system demineralization and Itai-Itai disease, with bone pain, physical impairment, and even bone fractures [162]. As Cd predominantly accumulates in the kidney long-term exposures to Cd can cause renal dysfunction, such as proteinuria, aminoaciduria, and decreased glomerular filtration rate [163]. Cd also presents toxicity to the cardiovascular system [164] and the reproductive system [165]. The treatments for the intoxication of Cd mainly include using chelating agents, such as ethylenediaminetetraacetic acid (EDTA), penicillamine (DPA), dimercaprol, and dithiocarbamates. Plasma exchange can also be used for Cd poisoning in emergency situations [166]. In 1993, Cd and its compounds were classified as Group 1 carcinogens by IARC according to the evidence from both human and experimental animal studies [4,6]. Cd exposure is associated with an increased risk of thyroid cancer, prostate cancer, breast cancer, bladder cancer, and lung cancer [167]. Cd has various cellular effects, such as impacting cell proliferation, apoptosis, and metabolism, and it may affect all stages of the carcinogenic process, including formation, progression, metastasis, and treatment response [168]. Studies have revealed some of the mechanisms of Cd-induced carcinogenesis, including the induction of oxidative stress, inhibited DNA repair machinery, aberrant gene expression, and perturbation of key signaling pathways involved in cancers [168,169]. Notably, although Cd is a well-established human carcinogen, the mechanisms by which it induces cancer are still poorly understood.

Cd is also a carcinogen with low affinity to DNA and low mutagenicity, suggesting that it may exert a tumorigenic effect through non-genotoxic ways [168]. In recent years, epigenetic mechanisms and non-coding RNAs have been shown to play important roles in Cd carcinogenesis. This section summarizes how epigenetic changes and non-coding RNAs are involved in Cd-induced carcinogenesis and angiogenesis. Unlike studies in Cr that largely focus on lung cancer, research on Cd-induced cancer is carried out in various types of cancer.

### 4.1. DNA Methylation

More and more studies have shown that Cd is able to modify DNA methylation status, causing both global and gene-specific chromatic hypermethylation and/or hypomethylation. A series of studies support that these alterations in DNA methylation and subsequent down-regulation of gene expression are involved in Cd carcinogenesis. Chronic exposure to low-dose Cd caused a low apoptosis rate, increased DNA DSBs, and caused global CpG island hyper- or hypomethylation in the liver of rats [170]. Cd exposure also led to reduced expression of caspase-8 due to promoter hypermethylation, leading to decreased hepatic apoptosis and increased preneoplastic lesions, evidenced by increased expression of cytokeratin 8/18 (a marker of liver preneoplastic lesions) [170]. Prolonged exposure to low-dose Cd resulted in the transformation of TRL 1215 liver cells, showing cancerous phenotypes such as hyperproliferation, high invasiveness, and decreased serum dependence in transformed cells [171,172,173]. The transformed cells exhibited significant increases in DNA methylation and DNA methyltransferase activity [171]. They also acquired high invasiveness, partially through down-regulation of apolipoprotein E (ApoE), an established suppressor of cell invasion, during malignant transformation, as re-expression of ApoE clearly abrogated the cell invasion in Cd-transformed cells [174]. The suppression of ApoE has been attributed to hypermethylation of the regulatory region of the ApoE promoter and Cd-mediated epigenetic suppression of liver X receptor α (LXRα), a transcriptional regulator for ApoE [174,175]. Down-modulation of TET1, a DNA demethylase in response to Cd-induced oxidative stress, might also cause down-regulation of ApoE via hypermethylation of the ApoE promoter [173].

Cd has been shown to act as a metalloestrogenic carcinogen and be associated with breast carcinogenesis [176,177,178], and aberrant epigenetic modifications are involved in Cd-induced breast cancers. Benbrahim-Tallaa et al. showed that chronic exposure to low-dose Cd led to the transformation of the immortalized human breast epithelial cell line MCF-10A, with the acquisition of typical cancer phenotypes, such as loss of contact inhibition, increased colony formation, and ability to form xenograft tumors in mice. The transformed cells displayed characteristics of basal-like breast carcinoma and harbored global DNA hypomethylation and c-Myc and K-Ras overexpression [176]. Liang et al. investigated the differential epigenome and transcriptome caused by Cd in MCF-7 breast cancer cells. They identified 997 Cd-induced differential genes potentially regulated by epigenetic mechanisms, and 400 of them were further validated in a large-scale breast cancer cohort, among which thioredoxin reductase 1 (TXNRD1) and chaperonin-containing TCP1 subunit 3 (CCT3) were identified as the critical genes that might play an important role in Cd-induced carcinogenesis [179]. Bioinformatics analyses also suggested that the epigenetically regulated differential genes were involved in important signaling pathways, such as Wnt signaling, metabolism, and MAPK signaling [179].

Most errors occurring during DNA replication can be corrected by the proofreading functions of DNA polymerase or by the post-replication MMR system. Inactivation of both error-proof systems by mutations and epigenetic changes may lead to the accumulation of a large number of mutations in cells and finally cause cancer [180]. Therefore, identifying epigenetic factors that inactivate the mutation-avoidance system has important implications for understanding the mechanisms for Cd-induced carcinogenesis. During Cd-induced malignant transformation of 16HBE cells, global DNA methylation was progressively increased, accompanied by progressively reduced expression of four DNA repair genes (mutS homolog 2 (hMSH2), ERCC1, XRCC1, and hOGG1) and the accumulation of DNA damage. Furthermore, the increased DNA methylation was associated with the overexpression of DNMT1 and DNMT3a; the reduced expression of hMSH2, ERCC1, XRCC1, and hOGG1 was likely caused by heavily methylated promoter regions of these genes. These results suggest that promoter hypermethylation-induced silencing of the DNA repair genes represents a potential mechanism underlying Cd-mediated carcinogenesis [181]. Cartularo et al. found that MGMT was epigenetically silenced in Cd-transformed BEAS-2B clones, which were more sensitive to an alkylating agent, temozolomide [182].

Hypermethylation-induced down-regulation of tumor suppressors has been shown to result in tumorigenesis and cancer development. During the malignant transformation of prostate epithelial RWPE-1 cells by Cd, a progressive increase in global DNA methylation associated with the overexpression of DNMT3b was observed. Meanwhile, the expression of two tumor-suppressor genes, RASSF1A and p16, was markedly reduced likely due to hypermethylation of the promoter regions [183]. Cd treatment promoted the growth of uveal and cutaneous melanoma cells due to the markedly reduced expression of p16INK4A and caspase-8 in the uveal and cutaneous melanoma cells, respectively. The silencing of p16INK4A and caspase-8 was attributed to hypermethylation induced by the increased activity of DNMTs [184]. Chronic low-dose Cd exposure has been shown to enhance lymphoblast proliferation in vitro and in vivo [185,186]. One mechanism for this phenomenon involved epigenetic silencing of tumor-suppressor gene p16. Yuan et al. showed that chronic low-dose Cd stimulated human B lymphoblast HMy2.CIR cell proliferation by down-regulating p16 via hypermethylation of CpG island in its promoter, suggesting epigenetic silencing of p16 contributed to Cd-induced carcinogenesis [187]. These studies indicate that genomic hypermethylation-induced quiescence of tumor suppressors may contribute to Cd-induced cancers.

Conversely, several studies have shown that Cd could induce global DNA hypomethylation. Huang et al. showed that Cd stimulated K562 lymphoblast proliferation and induced global DNA hypomethylation. The reduced global DNA methylation potentially contributed to Cd-stimulated K562 cell proliferation, because methionine, a donator of the methyl group in DNA methylation, was shown to prevent Cd-induced global DNA hypomethylation and Cd-stimulated cell proliferation [188]. Pelch et al. investigated the mRNA expression level and the promoter methylation status of five key genes relevant to the carcinogenic process (including S100 calcium-binding protein P (S100P), hyaluronidase 1 (HYAL1), neurotrimin (NTM), nestin (NES), aldehyde dehydrogenase 1 family member A1 (ALDH1A1)) in RWPE-1 cells malignantly transformed by Cd or As. Expressions of HYAL1 (hyaluronan degradation) and S100P (cancer aggressiveness) were up-regulated in transformed cells, correlated with hypomethylation of the chromatin near the transcriptional start sites of the two genes. In contrast, expressions of both NTM (cell adhesion) and NES (stem cell function) were decreased in the transformants, correlated with hypermethylation near the transcriptional start site. These results suggest that Cd- or As-induced alterations in DNA methylation may contribute to metal carcinogenesis via regulating the expression of important genes [189]. Cd exposure increased the viability of HepG2 and MCF7 cells. Cd also induced global DNA hypomethylation by lowering the levels of DNMT1, DNMT3A, and DNMT3B. Specifically, Cd-mediated DNMT down-regulation resulted in hypomethylation of the promoter of two known oncogenic methyltransferases, protein arginine methyltransferase 5 (PRMT5) and EZH2 [190,191]. This allows the enrichment of the nuclear transcription factor Y subunit alpha (NFYA) and E2F transcription factor-1 (E2F1), transcription factors in the PRMT5 and EZH2 promoters, respectively, to induce the expression of PRMT5 and EZH2. Up-regulated PRMT5 and EZH2, in turn, led to the increased global level of symmetric dimethylarginine (SDMA) and two crucial repressive histone marks, H4R3me2 and H3K27me3, which potentially silenced tumor suppressors through the remodeling of the chromatin. These results provide mechanistic insights into the DNA hypomethylation in Cd-induced cell proliferation and potential carcinogenesis [192]. The roles of aberrant DNA methylation/demethylation in Cd-induced carcinogenesis and cancer progression are summarized in Table 7.

### 4.2. Histone Modification

During the Cd-induced transformation of cells, multiple types of histone modifications have been discovered, which may contribute to Cd carcinogenesis through different mechanisms. For example, Liang et al. found that Cd-transformed BEAS-2B cells showed marked down-regulation of H3K4me2 and H3K36me3 and up-regulation of H3K9acS10ph, H4K5ac, H4K8ac, and H4K12ac histone modification marks [193]. The transformed cells also exhibited EMT and enhanced migration ability, while treatment of Cd-transformed cells with C646, a potent histone acetyltransferase inhibitor, suppressed the expression of mesenchymal marker genes and cell migration ability of these cells, suggesting that Cd-induced aberrant histone hyperacetylation is involved in EMT and lung cell transformation [193]. Cd exposure significantly promoted the proliferation, migration, and invasion of MCF-7 and T47-D breast cancer cells by inhibiting autophagy-related 5 (ATG5)-dependent autophagic flux. Mechanistically, a Cd-induced decrease in acyl-CoA synthetase short-chain family member 2 (ACSS2) expression inhibited ATG5 expression by reducing the level of H3K27ac in the promoter region of ATG5 [194]. Somji et al. found that metallothionein 3 (MT-3) expression was induced in UROtsa human urothelial cells transformed by Cd or As. H4 acetylation and methylation of H3K4, which are associated with transcriptional activation [16,195], were increased in the MT-3 promoter in the Cd- or As-transformed cells, while histone H3K9 and H3K27 methylation, which are associated with a transcriptionally repressed state, were also up-regulated in transformed cells. The pattern of histone modifications indicated that the MT-3 promoter in transformed cells had a “transcription ready” and “transcription repressed” bivalent chromatin structure compared with the parental cells, which allowed the metal-responsive transformation factor-1 (MTF-1) binding to metal response elements (MRE) of the MT-3 promoter more readily in the transformed cells to up-regulate MT-3 expression; however, this process was restricted in parental UROtsa cells [196]. Histone methylation, another type of histone modification, is also involved in Cd carcinogenesis. In BEAS-2B cells, exposure to Cd for 24 h led to increased global H3K4me3 and H3K9me2 by inhibiting the activities of H3K4 and H3K9 demethylases, respectively. Additionally, global H3K4me3 and H3K9me2 were also significantly increased at 4 weeks, whereas no significant change was observed from 8 to 20 weeks when the cells were transformed, suggesting that increased global H3K4me3 and H3K9me2 are involved in early events of Cd carcinogenesis [197]. The roles of histone modifications in Cd-induced carcinogenesis and cancer progression are summarized in Table 8.

### 4.3. Non-Coding RNAs

#### 4.3.1. miRNAs

A growing number of studies have shown that chronic exposure to Cd can cause fundamental changes in miRNA expression, thus mediating alterations in tumor-associated biological processes and facilitating the malignant transformation of cells. Yang et al. found that exposure to high-dose Cd resulted in cell viability reduction and apoptosis in IEC-6 intestinal epithelial cells. Cd exposure also caused global changes in miRNA expression, including up-regulated miR-124-3p and miR-370-3p. Elevated miR-124-3p and miR-370-3p, in turn, promoted Cd-induced apoptosis by directly targeting Bcl-2, a suppressor of apoptosis [198]. In another study, Cd was shown to induce apoptosis and ferroptosis, a type of programmed cell death dependent on iron, in PC12 rat pheochromocytoma cells. The mechanistic study indicated that Cd exposure induced miR-34a-5p expression, and up-regulated miR-34a-5p contributed to Cd-induced apoptosis and ferroptosis, likely through directly targeting sirtuin 1 (Sirt1), as the down-regulation of miR-34a-5p mitigated the apoptosis and ferroptosis caused by Cd. However, the role of down-regulated Sirt1 in miR-34a-5p-mediated apoptosis and ferroptosis was not validated in this study [199]. In BEAS-2B and BEP2D lung epithelial cells, Cd exposure resulted in the down-regulation of miR-30e, causing up-regulated expression of snail family transcriptional repressor 1 (SNAIL1), a direct target of miR-30e, and resultant EMT induction. EMT has been shown to play an important role in Cd-associated diseases such as fibrosis, COPD, and cancers [200]. Therefore, miR-30e down-regulation and the consequent activation in SNAIL1 expression might contribute to Cd-induced cancer formation and development through induction of EMT, although evidence is lacking for a causal relationship [201]. Additionally, Urani et al. reported down-regulated miR-34a and miR-200a and up-regulated SNAIL1 was induced by Cd in HepG2 cells [202]. Both miR-34a and miR-200a are known negative regulators for EMT in tumors [203,204], and SNAIL1 is a master regulator of EMT [205]; therefore, these results suggest that Cd may cause liver cancer by inducing EMT. Exposure to multiple metal carcinogens simultaneously is common in industrial settings. Short-term treatment of BALB/3T3 mouse fibroblasts with a mixture of As, Pb, and Cd could cause fundamental changes in miRNA expression patterns; many altered miRNAs were involved in various cellular functions, such as miR-154, miR-379, miR-204, and miR-133 [206]. Chronic exposure to the As-Cd-Pb mixture has been shown to induce malignant transformation of BALB/3T3 cells and increase miR-222 expression. Elevated miR-222 directly down-regulated Rad51c expression and impaired homologous recombination of DNA during the initiation stage of cell transformation, suggesting that miR-222 is an initiator of Cd-induced carcinogenesis [207].

It has been well-known that higher levels of Cd can cause vascular toxicity by primarily targeting endothelial cells. Angiogenesis is a critical step for tumor growth and metastasis. However, the effect of Cd on tumor angiogenesis has been poorly studied. Current studies indicate that Cd has a dose-dependent effect on tumor angiogenesis. Low doses (1 μM-10 μM) of Cd have been shown to promote tumor angiogenesis [208,209,210,211] by activating PKB/Akt [212], NF-κB [213], and MAPKs [210] signaling pathways, resulting in endothelial cell activation and tumor angiogenesis [214]; high concentrations (>10 μM) of Cd may cause damage and apoptosis of endothelial cells [210,215] and decrease tumor angiogenesis [210]. In a study by Che et al., Cd was shown to induce cytotoxicity and promote the tube-formation ability of primary human umbilical vein endothelial cells (HUVECs) by down-regulating miR-101. Suppressed miR-101, in turn, induced COX2 expression and endoplasmic reticulum (ER) stress; up-regulated COX2 and ER stress led to increased vascular endothelial growth factor (VEGF) protein level, resulting in abnormal angiogenesis [216]. Notably, the roles of epigenetic modifications and non-coding RNAs in mediating the effect of Cd on angiogenesis remain largely unclear; further studies are necessarily needed. The roles of altered miRNAs in Cd-caused cancers are summarized in Table 9.

#### 4.3.2. LncRNA

In recent years, the contributions of lncRNAs in Cd carcinogenesis have been extensively investigated. Here, we summarize the up-to-date advances in studying the roles of lncRNAs in Cd-mediated tumorigenesis.

Cell growth and apoptosis play important roles in cancer formation and development. Some studies have indicated that lncRNAs contribute to Cd carcinogenesis by regulating cell proliferation and apoptosis. Exposure to Cd is related to the increased occurrence of prostate cancer, and workers occupationally exposed to Cd have a higher risk of mortality from prostate cancer [220,221]. Additionally, chronic exposure to Cd promoted the malignant transformation of the normal prostate epithelial (PWR1E and RWPE1) cells, suggesting a causative role of Cd in prostate cancer [222]. Long-term exposure of PC3 and DU145 prostate cancer cells to low-dose Cd has been shown to promote cell growth and induce resistance to ferroptosis in vitro and in vivo [217]. LncRNA OIP5-AS1 expression was greatly up-regulated in Cd-exposed PC3 and DU145 cells; up-regulated OIP5-AS1 served as an endogenous sponge of miR-128-3p to up-regulate the expression of SLC7A11, a direct target of miR-128-3p, thereby inhibiting ferroptosis [217]. LncRNA-MALAT1 expression was up-regulated in Cd-transformed 16HBE cells, the lung tissues of Cd-exposed rats, and the blood of Cr-exposed workers; furthermore, the blood MALAT1 expression had a positive correlation with urinary/blood Cd concentrations in the workers. Silencing MALAT1 inhibited cell proliferation and cell cycle progression, migration, and invasion, and induced apoptosis in Cd-transformed 16HBE cells. The expression of forkhead box C2 (FOXC2), STAT3, BCL2-associated X, apoptosis regulator (BAX), EGFR, and TGF-β1 was reduced, but Bcl-2 was increased in MALAT1-depleted cells, Additionally, there were positive correlations of MALAT1 with the expressions of target genes, such as STAT3, BAX, and TGF-β1, in the lungs of Cd-exposed rats and the blood of Cd-exposed workers. These results suggest that MALAT1 contributes to Cd-mediated carcinogenesis and tumor progression by regulating key genes related to cell proliferation, apoptosis, migration, and invasion [223].

Cd-induced carcinogenesis is partly due to the accumulation of DNA damage and chromosomal aberrations. In recent years, lncRNAs have been shown to be involved in modulating DNA damage and repair in Cd toxicology. The Cd-transformed 16HBE cells had a large number of lncRNAs with altered expression levels [224,225]. Among the dysregulated lncRNAs, ENST00000414355 [224] and ENST00000446135 [225] were significantly up-regulated in transformed 16HBE cells, the lung of Cd-exposed rats, and the blood of Cd-exposed workers. Silencing of ENST00000414355 [224] and ENST00000446135 [225] both reduced DNA damage and the expressions of DNA damage-related genes (ATM serine/threonine kinase (ATM), ATR serine/threonine kinase (ATR), and ATR interacting protein (ATRIP)) but increased the expressions of DNA repair-related genes (damage-specific DNA binding protein 1 (DDB1), DDB2, OGG1, ERCC1, MSH2, RAD50, XRCC1, and BRCA1-associated RING domain 1 (BARD1)), suggesting that these two lncRNAs contribute to Cd carcinogenesis by causing the accumulation of DNA damage and inhibiting DNA repair systems. Additionally, ENST00000446135 could directly regulate the transcription of MSH2, a DNA repair-related gene [225]. LncRNA MT1DP has been shown to promote Cd-induced DNA damage response, genome instability, and DNA replication stress, and it inhibits homologous recombination repair in HepG2 cells. Mechanically, ATR was activated to enhance HIF-1α expression upon Cd exposure, which in turn promoted the transcription level of MT1DP. Up-regulated MT1DP was then recruited on the chromatin and bonded to SMARCAL1 (SWI/SNF-related, matrix-associated, actin-dependent regulator of chromatin, subfamily a-like 1) to competitively inhibit the interaction of SMARCAL1 with the RPA complexes, finally leading to increased replication stress and DNA damage. Therefore, MT1DP contributed to Cd carcinogenesis via regulating Cd-induced DNA damage and replication stress by inhibiting the recruitment of SMARCAL1 to chromatin [226].

Cd is a well-known lung carcinogen; long-term exposure to low dose Cd has been shown to cause BEAS-2B cell malignant transformation and obtaining of cancer stem cell (CSC)-like phenotypes [227,228]. MEG3, a tumor-suppressive lncRNA, was significantly down-regulated in Cd-transformed BEAS-2B cells likely due to the increased methylation of the differentially methylated region (DMR) upstream of the MEG3 transcription start site by induction of DNMTs. The reduced expression of MEG3 promoted Cd-exposure-induced cell transformation and acquisition of CSC-like properties. Forced expression of MEG3 could reduce cell proliferation, cell cycle progression, and colony formation in soft agar assay; cause resistance to Cd-induced apoptosis; and lower the number of colonies in the serum-free suspension culture assay, an indicator of the stemness of the cells. Further mechanistic studies revealed that forced expression of MEG3 led to increased p21 levels, reduced Rb phosphorylation, and B-cell lymphoma extra large (Bcl-xL) levels in Cd-exposed cells, thus reducing cell cycle progression and apoptosis resistance. These findings suggest that suppressed MEG3 expression contributes to Cd-induced cell transformation and acquisition of CSC-like properties by promoting cell cycle progression and apoptosis resistance [227]. Lin et al. found significant up-regulation of an oncogenic lncRNA DUXAP10 in Cd-transformed cells, and that elevated DUXAP10 plays an important role in Cd-induced cell transformation and CSC-like property; DUXAP10 depletion in Cd-transformed cells resulted in significantly decreased stemness markers (including KLF transcription factor-4 (KLF4), KLF5, Nanog, and surface CD133) and suspension spheres number. Mechanistically, Cd transformation led to an increased level of paired box 6 (PAX6), which transcriptionally up-regulated the expression of DUXAP10. Elevated DUXAP10, in turn, activated the hedgehog signaling pathway, which is essential in maintaining the CSC-like properties in Cd-transformed cells, by increasing glioma-associated oncogene homolog 1 (GLI1) protein stability [228]. The above-mentioned studies highlight the important roles of lncRNAs in Cd exposure-induced tumorigenesis and CSC-like properties.

#### 4.3.3. CircRNAs

CircRNA is a recently identified non-coding RNA type, usually more than 200 nt in length. CircRNAs have a ringlike structure, unlike lncRNAs, which are linear. CircRNAs are transcribed from exon, intron, or intergenic regions and folded into complicated second structures [229]. CircRNAs have been shown to regulate gene expression through multiple mechanisms. They can act as miRNA decoys to prevent miRNAs from binding to their targets and protect the target transcripts from degradation [229]. They can interfere with transcription factors to bind to promoters and thus regulate targeted gene expression [229]. They can also work as a scaffold to regulate protein-protein interactions and the related downstream signaling pathways [230]. Recently, some studies showed that lncRNAs and circRNAs also participate in the epigenetic modulation of chromatin to regulate gene expression [231]. A growing number of studies demonstrate that circRNAs act as either tumor suppressors [232] or oncogenes [233] in cancer development. In recent years, an increasing number of studies have shown that circRNA expression is disturbed upon exposure to Cd and in Cd-transformed cells, and dysregulated circRNAs are involved in Cd-induced cancers. For example, several studies have indicated that circRNAs are implicated in Cd carcinogenesis via sponging microRNAs to regulate gene expression. Circ-SHPRH was down-regulated during Cd-induced transformation of BEAS-2B cells, and suppression of circ-SHPRH played a promoting role in Cd-carcinogenesis; forced expression of circ-SHPRH inhibited cell proliferation, EMT, migration, invasion, and anchorage-independent growth in transformed BEAS-2B cells and prevented the Cd-induced transformation of BEAS-2B cells [218]. Mechanistically, circ-SHPRH functioned as a sponge of miR-224-5p; suppressed circ-SHPRH led to up-regulated miR-224-5p and resultant down-regulation of QKI (QKI, KH domain containing RNA binding), a direct target of miR-224-5p and a tumor-suppressor protein known to prevent proliferation and EMT during the progression of human cancers [234,235], promoting Cd-induced tumorigenesis [218]. CircSPAG16 was also significantly down-regulated in Cd-transformed BEAS-2B cells; overexpression of circSPAG16 inhibited the hallmarks of cancer in Cd-transformed cells, such as cell proliferation, migration, invasion, and anchorage-independent growth. CircSPAG16 overexpression also prevented the Cd-induced transformation of BEAS-2B cells. Mechanistically, circSPAG16 inhibited Cd carcinogenesis by suppressing oncogenic phosphatidylinositol 4-phosphate 5-kinase type-1 α (PIP5K1α) via direct interaction; suppressed PIP5K1α led to the inactivation of Akt. Therefore, down-regulated circSPAG16 contributed to Cd-induced carcinogenesis by promoting PIP5K1α-induced activation of Akt [236]. Another circRNA, circPUS7, has been shown to be up-regulated during the Cd-induced transformation of BEAS-2B cells; elevated circPUS7 contributed to Cd-induced transformation of BEAS-2B cells by sponging miR-770, resulting in up-regulated K-Ras [237], an oncogene playing important roles in cell proliferation and transformation [238]. Furthermore, the knockdown of circPUS7 in transformed BEAS-2B cells significantly attenuated cell proliferation, migration, invasion, and anchorage-independent growth [237]. These results indicate that circPUS7 plays an important role in Cd-tumorigenesis and in maintaining the cancer phenotypes of Cd-transformed cells. Unlike the functions in lung cancer, Cd has been shown to induce apoptosis and inflammatory reactions in bovine mammary epithelial cells (BMECs) and in mouse mammary glands [239]. Circ08409 was up-regulated in Cd-exposed cells; elevated circ08409 promoted apoptosis and led to altered expression of a series of inflammatory factors in BMECs through directly sponging miR-133a, a miRNA directly targeting TGFB2, to up-regulate TGFB2, which modulates cell proliferation, apoptosis, and inflammation. The data highlighted the importance of the circ08409/miR-133a/TGFB2 axis in mediating the pro-apoptotic and pro-inflammatory effects of Cd in mammary tissues [239].

#### 4.3.4. RNA N6-Methyladenosine (m6A) Modification

Epigenetic modifications of RNAs, including mRNA, miRNA, and lncRNA, represent another layer of epigenetic regulation of gene expression [240]. The m6A modification is the most common post-transcriptional modification of RNA molecules that regulates RNA metabolism; more and more studies have indicated that m6A-mediated gene expression is involved in the development of various cancers [241]. However, its role in metal carcinogenesis has rarely been investigated. Li et al. showed that the global level of mRNA m6A modification was significantly decreased in Cd-transformed BEAS-2B cells, mediated by Cd-induced up-regulation of alkB homolog 5, RNA demethylase (ALKBH5), an m6A demethylase [242]. Furthermore, ALKBH5 promoted the proliferation, migration, invasion, and anchorage-independent growth of transformed BEAS-2B cells by reducing the m6A level of phosphatase and tensin homolog (PTEN) mRNA via demethylating m6A in PTEN mRNA, resulting in its instability and the reduction in PTEN protein expression [242]. These results suggest that ALKBH5 promotes Cd-induced carcinogenesis by reducing PTEN mRNA stability in an m6A-dependent manner [242]. In Cd-transformed SV-HUC-1 human uroepithelial cells, Cd caused global changes in mRNA and protein expression and m6A-modified genes. The integrated analysis of the transcriptome, proteome, and m6A profiles revealed that some of the shared genes were involved in DNA damage stimulus, cell proliferation, and onset or progression of cancer; this suggests that m6A modification-mediated post-transcriptional regulation may be involved in Cd-induced malignant transformation of uroepithelial cells [243]. In another study, Yue et al. showed that reduced m6A modification of miR-374c-5p contributed to Cd-induced progression of breast cancer [219]. In T-47D and MCF-7 breast cancer cells, Cd exposure caused reduced m6A modification of pri-miRNA-374c, resulting in miR-374c-5p down-regulation by decreasing pri-miRNA-374c transcript stability. Suppression of miR-374c-5p, in turn, promoted proliferation, migration, and invasion of cells by activating glutamate metabotropic receptor 3 (GRM3) expression [219], which is a glutamate receptor modulating glutamate processing and has been shown to drive breast cancer cell metastasis [244].

## 5. Conclusions and Future Perspectives

Environmental and occupational exposure to heavy metals, such as Cr(VI), Ni, and Cd, remain serious public health concerns that harm hundreds of millions of people globally, causing diseases and cancers in the digestive, respiratory, and urinary systems. Therefore, a complete understanding of the mechanisms of metal carcinogenesis is important in preventing and treating heavy metal-induced cancers. In recent years, a growing number of studies have shown that short- or long-term exposure to Cr(VI), Ni, or Cd caused global changes in epigenetic modifications and non-coding RNA expression in cells. Epigenetic modifications and non-coding RNAs are important regulators of gene expression and have been recognized as important players in tumor formation, development, and metastasis. The currently available findings from studies discussed in this review indicate that altered epigenetic modifications and dysregulated expression of non-coding RNAs (particularly miRNA) are critically involved in metal carcinogen-induced cell malignant transformation, tumorigenesis, and angiogenesis. Notably, although the number of studies is fast growing over the past decade, only a small part of the hundreds of epigenetic changes and altered non-coding RNAs from heavy metal exposure has been characterized and the significance of most of these changes is still unknown. Therefore, to support the diagnosis and treatment of human cancers resulting from metal carcinogen exposure, further studies are required to better define the role of epigenetic modifications and miRNA in heavy metal carcinogenesis and angiogenesis.

We should also acknowledge the limits of the current studies. Firstly, most studies were carried out in cultured cells, such as 16HEBs and BEAS-2B cells, or xenograft models based on transformed cells, which may not fully reflect the roles of epigenetic modifications and non-coding RNAs in heavy metal exposure-induced cancers in vivo. More studies using animal models such as mice and rats need to be performed to further reveal the important role of dysregulation of epigenetic modifications and non-coding RNA in metal carcinogenesis. Second, most of the knowledge about the roles of certain epigenetic markers or non-coding RNAs is based on altering their levels in cells already transformed by metal carcinogens to investigate the changes in malignant phenotypes. However, fewer studies have been performed to investigate non-transformed cells to investigate if they are able to prevent or promote metal-induced cell transformation. More studies based on non-transformed cells or genetically modified animal models should be performed to more accurately reveal the roles of dysregulated epigenetic marks and non-coding RNAs in metal carcinogen-caused tumorigenesis. Finally, the knowledge of the roles of epigenetic modifications and non-coding RNAs in metal carcinogen-induced cancer should be translated into clinical work. More studies in animals and humans are needed to evaluate their roles as biomarkers for early diagnosis and to develop preventive methods and treatment options for metal carcinogen exposure-induced human cancers in the future.

## Figures and Tables

**Figure 1 cancers-14-05768-f001:**
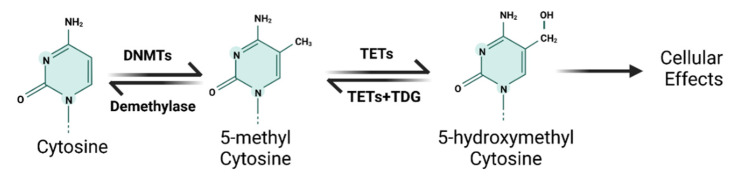
The schematic diagram of DNA methylation and demethylation mechanism.

**Figure 2 cancers-14-05768-f002:**
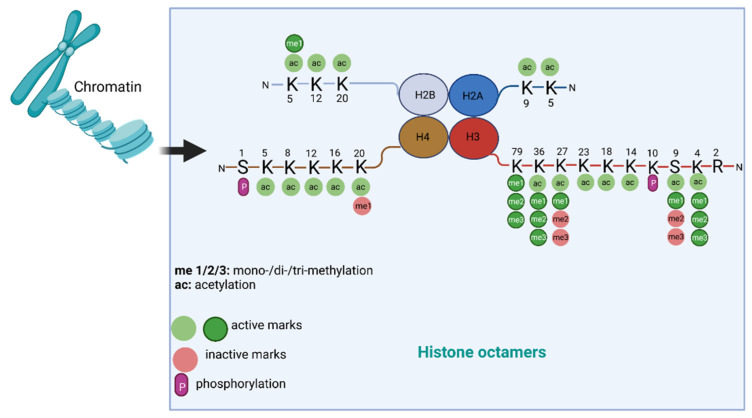
Common histone modifications and their functions.

**Figure 3 cancers-14-05768-f003:**
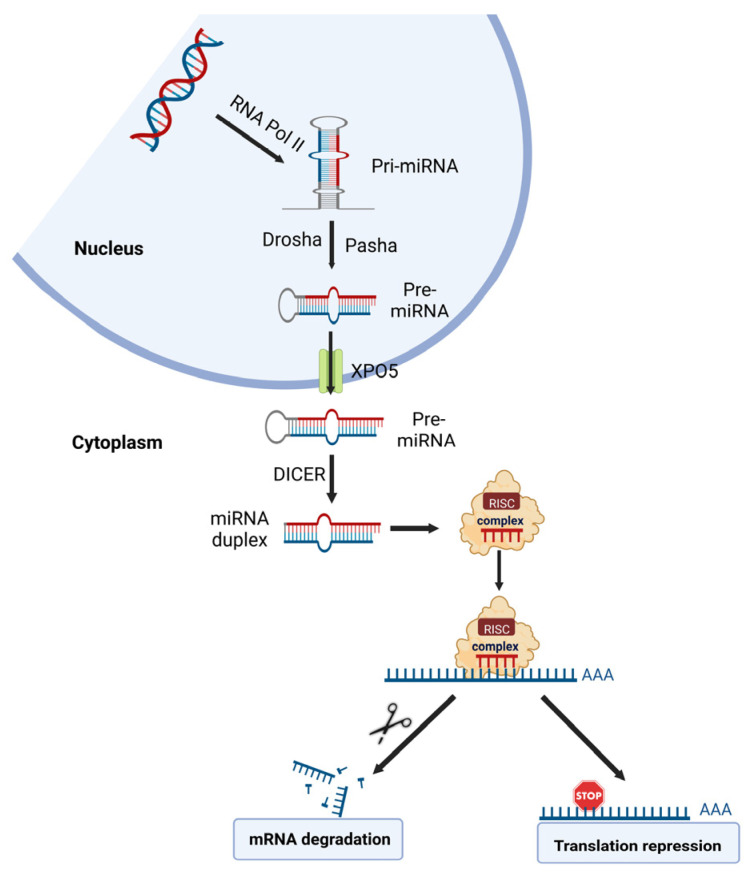
The schematic diagram of microRNA biogenesis and roles in gene expression.

**Table 1 cancers-14-05768-t001:** Roles of altered DNA methylation in Cr(VI) exposure-induced carcinogenesis.

Cells or Tissues	Compound	Route	Exposure Time	Change (s)	Role	Mechanism
16HBE cells [50]	Dichromate (Cr_2_O_7_^2−^)	Culture medium	24 h	Hypermethylation at CpG1, CpG31, and CpG32 of p16INK4a promoter	Pro-oncogenic	Hypermethylation of the promoter caused lowered p16INK4a expression and accumulation of DNA damage.
Human lung cancer [51]	Chromate	Inhalation	>15 y			
16HBE/human lung cancer [52]	Potassium dichromate (K_2_Cr_2_O_7_)	Culture medium/ inhalation	24 h	Hypermethylation of MGMT, HOGG1, RAD51, XRCC1, and ERCC3	Pro-oncogenic	Decreased the expression of these DNA repair genes, leading to suppressed DNA repair system and resultant accumulation of genetic damages.
Human lung cancer [54,56]	Chromate	Inhalation	N/A	Hypermethylation of MLH1 promoter	Pro-oncogenic	Inhibit the expression of MLH1, a DNA mismatch repair gene, causing increased microsatellite instability of the genome.
Human lung cancer [57]	Chromate	Inhalation	N/A	Global DNA hypermethylation Hypermethylation of p16 and APC promoters	Pro-oncogenic	Globally increased genomic methylation; increased promoter methylation caused decreased expression of p16 and APC, two tumor suppressors, contributing to Cr(VI) carcinogenesis.

Cr(VI): hexavalent chromium; MGMT: O6-methylguanine-DNA-methyltransferase; HOGG1: 8-oxoguanine DNA glycosylase; RAD51: RAD51 recombinase; XRCC1: X-ray repair cross-complementing 1; ERCC3: ERCC excision repair 3; MLH1: MutL homolog 1; APC: adenomatous polyposis coli.

**Table 2 cancers-14-05768-t002:** Roles of histone modifications in Cr(VI) exposure-induced carcinogenesis.

Cells or Tissues	Compound	Route	Exposure Time	Modification (s)	Role	Mechanism
BEAS-2B/16HBE cells [46]	K_2_Cr_2_O_7_	Culture medium	20 w/40 w	Increased global H3K9me2 and H3K27me3	Pro-oncogenic	Cr(VI) led to up-regulated global H3K9me2 and H3K27me3 by increasing the levels of HMTs. The aberrant histone methylation contributed to Cr(VI)-induced genotoxicity and carcinogenesis.
Human lung cancer [57]	Chromate	Inhalation	NA	Globally increased H3K9me2/3 and H3K4me2/3, but decreased H3K27me3 and H3R2me2 Increased H3K9me2 in MLH1 promoter	Pro-oncogenic	Cr(VI) induced globally increased H3K9me2/3 and H3K4me2/3, but decreased H3K27me3 and H3R2me2. Cr(VI) also caused increased H3K9me2 in MLH1 promoter, likely via up-regulating G9a and SUV39H1, causing repressed expression of MLH1, a tumor suppressor.
A549 [58]	K_2_CrO_4_	Culture medium	1 h–24 h			
A549 [59]	K_2_CrO_4_	Culture medium	24 h			
BEAS-2B [60]	K_2_CrO_4_	Culture medium	6 months	Reduced H3K9ac and H3K4me3, but enriched H3K27me3 in the promoter of HHIP	Pro-oncogenic	Cr(VI) caused reduced levels of H3K9ac and H3K4me3 but enriched H3K27me3 in the promoter of HHIP, leading to decreased expression of HHIP. Decreased HHIP activated the hedgehog signaling, which contributed to Cr(VI) carcinogenesis.
BEAS-2B [62]	K_2_Cr_2_O_7_	Culture medium	6 months	Enhanced histone H3 acetylation in the promoter of CXCL5	Pro-oncogenic	Cr(VI) activated the c-Myc/p300 complex, which was specifically bound to the CXCL5 promoter and enhanced H3 acetylation and eventually promoted CXCL5 transcription. Up-regulated CXCL5 contributed to Cr(VI) carcinogenesis.
BEAS-2B [63]	K_2_CrO_4_	Culture medium	24 h/1–2 w	Increased H3K9ac and H3K14ac of NUPR1 promoter	Pro-oncogenic	Cr(VI) caused increased H3K9ac and H3K14ac of NUPR1 promoter and elevated NUPR1 expression, which in turn promoted Cr(VI)-induced transformation of BEAS-2B cells.
BEAS-2B cells [69]	K2Cr2O7	Culture medium	20 w	Increased H3K9ac and H3K27ac at the promoter of ACLY	Pro-oncogenic	Cr(VI) led to a glycolytic shift, which led to increased H3K9ac, H3K27ac, and H2B and H4 acetylation. The up-regulated H3ac at the ACLY promoter promoted ACLY transcription; up-regulated ACLY in turn increased the glycolytic shift and histone acetylation, forming a positive feedback loop. The glycolytic shift played a critical role in maintaining the malignant phenotypes of Cr(VI)-transformed cells.
16HBE [70,71]	K_2_CrO_4_	Culture medium	15 w	Global and 53BP1 promoter-specific decreases in H3K18ac and H3K27ac	Pro-oncogenic	Cr(VI) caused increased SET, resulting in global and 53BP1 promoter-specific reduction in H3K18ac and H3K27ac. The promoter histone hypoacetylation resulted in decreased expression of 53BP1, which contributed to Cr(VI) carcinogenesis by causing DNA damage accumulation and inhibition of apoptosis.

Cr(VI): hexavalent chromium; HMTs: histone-lysing methyltransferases; MLH1: MutL homolog 1; HHIP: hedgehog-interacting protein; CXCL5: C-X-C motif chemokine ligand 5; NUPR1: nuclear protein 1; ACLY: ATP citrate lyase; SET: SET nuclear proto-oncogene; 53BP1: P53 binding protein 1.

**Table 3 cancers-14-05768-t003:** Roles of altered miRNA in Cr(VI) exposure-induced carcinogenesis.

Cells or Tissues	Compound	Route	Exposure Time	miRNA	Alteration	Role	Target (s)	Mechanism
BEAS-2B/BALB/cJ mice [75]	Na_2_Cr_2_O_7_/ zinc chromate	Culture medium/intranasal instillation	6 months /12 w	miR-27a miR-27b	Down	Tumor suppressor	NRF2	MiR-27a/b were down-regulated in response to ROS production upon Cr(VI) exposure; down-regulated miR27a/b promoted Cr(VI)-induced tumorigenesis and angiogenesis by up-regulating NRF2.
BEAS-2B [77,78]	K_2_Cr_2_O_7_	Culture medium	24 h/ 6 months	miR-21	Up	Oncogenic	PDCD4	Cr(VI) promoted miR-21 transcription by activating STAT3. Up-regulated miR-21 directly targeted PDCD4, a tumor suppressor. Inhibition of PDCD4 contributed to Cr(VI)-induced cancer by suppressing the expression of E-cadherin, c-Myc, and uPAR.
L02 hepatocytes [81]	NA	Culture medium	24 h	miR-21	Down	Pro-apoptotic	PDCD4	Cr(VI) caused a decrease in miR-21 and a subsequent increase in PDCD4, contributing to Cr(VI)-induced apoptosis and decreased proliferation of liver cells.
BEAS-2B [83,84]	Na_2_Cr_2_O_7_	Culture medium	6 months	miR-143	Down	Tumor suppressor	IGF-IR IRS1	Cr(VI) lowered miR-143 expression; decreased miR-143 promoted Cr(VI)-induced tumor growth and angiogenesis by activating IGF-IR/IRS1/ERK, mTOR/p70S6K1, HIF-1α/VEGF, and NF-κB p65 pathways.
Human lung cancer [85]	Na_2_Cr_2_O_7_	Inhalation	3.0–10.0 years	miR-3940-5p	Down	Oncogenic	NA	Cr(VI) led to decreased miR-3940-5p; the lowered miR-3940-5p then enhanced the HR of DSBs to mitigate the accumulation of DNA damage, playing a protective role in Cr(VI)-caused cell transformation.
16HBE [86]	Na_2_CrO_4_	Culture medium	24 h					
BEAS-2B [88]	K_2_Cr_2_O_7_	Culture medium	20 w	miR-494	Down	Tumor suppressor	c-Myc	Cr(VI) caused down-regulation of miR-494, which in turn led to increased c-Myc, contributing to Cr(VI)-induced transformation and acquisition of CSC-like properties in BEAS-2B cells.

Cr(VI): hexavalent chromium; ROS: reactive oxygen species; Nrf2: NF-E2-related factor-2; PDCD4: programmed cell death 4; uPAR: plasminogen activator, urokinase receptor; IGF-IR: insulin-like growth factor-1 receptor; IRS1: insulin receptor substrate-1; HR: homologous recombination; DSBs: double-strand breaks; CSC: cancer stem cells.

**Table 4 cancers-14-05768-t004:** Roles of altered DNA methylation in Ni-induced carcinogenesis.

Cells or Tissues	Compound	Route	Exposure Time	Modification (s)	Role	Mechanism
16HBE cells [100]	Crystalline NiS	Culture medium	24 h	Hypermethylation of MGMT promoter	Pro-oncogenic	MGMT was down-regulated by DNA hypermethylation, reduced histone H4-ac and H3K9ac, and up-regulated H3K9me2. Lowered expression of MGMT contributed to Ni-induced malignant transformation.
BEAS-2B cells [102]	NiCl_2_	Culture medium	72 h/ 6–9 d	Hypermethylation of E-cadherin promoter	Pro-oncogenic	Ni inhibited E-cadherin expression by induction of ROS-dependent promoter hypermethylation, resulting in the acquisition of EMT, which contributed to Ni carcinogenesis.
Wistar rats [103]	Ni_3_S_2_	Intra-muscular injection	32 w	Hypermethylation of RAR-β2, RASSF1A, and CDKN2A	Unknown	Ni induced hypermethylation of the 5’ region of RAR-β2, RASSF1A, and CDKN2A, resulting in decreased mRNA expressions of these genes.
C57BL6 mice [104]	Ni_3_S_2_	Intra-muscular injection	8 months	Hypermethylation of p16Ink4a	Pro-oncogenic	Ni induced hypermethylation and down-regulated expression of p16Ink4a. The p16Ink4a silence together with activation of the MAPK signaling may contribute to Ni carcinogenesis.
Dermal fibroblasts of Syrian hamster [105]	NiCl_2_	Culture medium	N/A	Hypermethylation of p16Ink4a	Pro-oncogenic	Promoter hypermethylation-induced silencing of p16Ink4a contributed to Ni-induced immortalization of primary dermal fibroblast SHD cells.

Ni: nickel; MGMT: O-6-methylguanine-DNA methyltransferase; ROS: reactive oxygen species; EMT: epithelial-to-mesenchymal transition; RAR-β2: retinoic acid receptor beta; RASSF1A: Ras association domain family member 1; CDKN2A: cyclin-dependent kinase inhibitor 2A.

**Table 5 cancers-14-05768-t005:** Roles of altered histone modifications in Ni exposure-induced carcinogenesis.

Cells or Tissues	Compound	Route	Exposure Time	Modification (s)	Role	Mechanism
BEAS-2B cells [111]	NiCl_2_	Culture medium	6 w	Global increases in H3K4me3 and H3K27me3	Unknown	Ni caused genome-wide increases in H3K4me3 and H3K27me3, but the role of these changes was unknown.
BEAS-2B [112]	NiCl_2_	Culture medium	72 h	Global H3K9me2 spreading	Unknown	Ni caused disrupted H3K9me2 (a repressive mark) domains and subsequent spreading of H3K9me2 into active chromatin regions, causing global gene silencing.
A549 cells [59]	K_2_CrO_4_	Culture medium	24 h	Global increase in H3K9me2 and H3K4me3 Increased H3K4me3 of CA9 and NDRG1	Unknown	Ni led to increased global levels of H3K9me2 and H3K4me3. Increased H3K4me3 was also discovered in the promoter and coding regions of CA9 and NDRG1, which were up-regulated in Ni-exposed A549 cells.
A549 [113]	NiCl_2_					
A549 [114]	NiCl_2_					
BEAS-2B [115]	NiCl_2_	Culture medium	72 h/6 w	Decreased H3K27me3 at the ZEB1 promoter	Pro-oncogenic	Ni led to decreased H3K27me3 (a repressive mark) level at the ZEB1 promoter; loss of H3K27me3 likely caused increased ZEB1 expression, finally resulting in EMT in Ni-exposed cells.
BEAS-2B [116]	NiCl_2_	Culture medium	24 h/8 w	Increased H3K9me2 at the SPRY2 promoter	Pro-oncogenic	Ni caused increased H3K9me2 at the SPRY2 promoter by inhibiting JMJD1A, resulting in repressed SPRY2 expression. SPRY2 is a negative regulator of ERK signaling; repression of SPRY2 thus activated the ERK signaling to promote Ni carcinogenesis.
HEK293T/786-0 cells [123]	NiCl_2_	Culture medium	24 h	Reduced H3K27me3 level	Unknown	Ni caused a reduced level of H3K27me3, possibly caused by up-regulated JMJD3, an H3K27me3 demethylase, but the function of decreased H3K27me3 was not studied.
HEK293/A549 cells [125]	NiCl_2_	Culture medium	48 h	Decreased H3K4me2 of E-cadherin promoter	Pro-oncogenic	Ni caused recruitment of LSD1 complex to E-cadherin promoter to catalyze the reduction in H3K4me2 modification of E-cadherin promoter, resulting in down-regulated E-cadherin level and induction of EMT.
16HBE [100]	Crystalline NiS	Culture medium	24 h	Reduced H4ac and H3K9ac and up-regulated H3K9me2 of MGMT	Pro-oncogenic	MGMT was down-regulated by DNA hypermethylation, reduced histone H4ac, and H3K9ac and up-regulated H3K9me2. Lowered expression of MGMT led to impaired DNA repairing machinery.
Hep3B cells [132]	NiCl_2_	Culture medium	2–24 h	Global histone hypoacetylation Reduced H4ac in Bcl-2 promoter	Tumor suppressive	Ni induced histone hypoacetylation by inhibiting the overall HAT activity. Ni also led to reduced H4 acetylation in the Bcl-2 promoter, resulting in Bcl-2 down-regulation, which contributed to Ni-caused cell growth inhibition and apoptosis.
Hep3B [133]			12–48 h			
A549 [135]	NiCl_2_	Culture medium	24–48 h	Increased H3S10 phosphorylation	Unknown	Ni induce phosphorylation of H3S10 by activating the JNK/SAPK signaling pathway, but the role of H3S10 phosphorylation was not studied.

Ni: nickel; CA9: carbonic anhydrase 9; NDRG1: N-Myc downstream regulated 1; ZEB1: zinc finger E-box binding homeobox 1; EMT: epithelial-to-mesenchymal transition; SPRY2: sprouty RTK signaling antagonist 2; JMJD1A: Jumanji domain-containing protein 1A; LSD1: lysine-specific demethylase 1; MGMT: O-6-methylguanine-DNA methyltransferase; HAT: histone acetyltransferase; JNK: c-Jun N-terminal kinase; SAPK: stress-activated protein kinase.

**Table 6 cancers-14-05768-t006:** Roles of altered miRNA in Ni exposure-induced carcinogenesis.

Cells or Tissues	Compound	Route	Exposure Time	miRNA	Alteration	Role	Target (s)	Mechanism
BEAS-2B A549 cells [143]	NiCl_2_	Culture medium	15 min–72 h	miR-4417	Up	Oncogenic	TAB2	Activation of miR-4417/TAB2 was involved in Ni-induced EMT and carcinogenesis.
BEAS-2B [144]	NiSO_4_	Culture medium	4 w	miR-31	Down	Tumor suppressor	SATB2	RUNX2 transcriptionally inhibited miR-31, leading to up-regulation of SATB2 to contribute to Ni-induced BEAS-2B cell transformation.
H1355/ H23 cells [146]	NiCl_2_	Culture medium	24 h	miR-21	Up	Oncogenic	SPRY2 RECK	Activation of the EGFR/NF-κB signaling pathway upon Ni exposure induced miR-21, resulting in down-regulation of SPRY 2 and RECK to promote the invasiveness of lung cancer cells.
Wistar rats [147]	Ni_3_S_2_	Intra-muscular injection	32 w	miR-222	Up	Oncogenic	CDKN1B CDKN1C	In muscle tumors and lung cancer, miR-222 mediated the down-regulation of CDKN1B and CDKN1C, which contributed to Ni-induced tumorigenesis.
16HBE [148]	Crystalline Ni_3_S_2_	Culture medium	13 rounds of chronic exposure	miR-203	Down	Tumor suppressor	ABL1	Down-regulation of miR-203 resulted in the up-regulation of ABL1, contributing to Ni-induced cancer.
16HBE [149]	Crystalline NiS	Culture medium	NA	miR-152	Down	Tumor suppressor	DNMT1	Inhibition of miR-152 resulted in increased expression of DNMT1, which promotes Ni-induced cell growth and malignant transformation.
Neuro-2a cells [150]	NiCl_2_	Culture medium	2–8 h	miR-210	Up	Oncogenic	ISCU1/2	Up-regulated HIF-1α promoted the transcription of miR-210; the elevated miR-210 modulates the energy metabolism shift to aerobic glycolysis through targeting ISCU1/2.
BEAS-2B [156]	NiCl_2_	Culture medium	24 h					

Ni: nickel; TAB2: TGF-beta activated kinase 1 (MAP3K7) binding protein 2; SATB2: special AT-rich sequence-binding protein 2; SPRY2: sprouty RTK signaling antagonist 2; RECK: reversion-inducing cysteine-rich protein with kazal motifs; CDKN1B: cyclin-dependent kinase inhibitor 1B; ABL1: ABL proto-oncogene 1; ISCU1/2: iron–sulfur cluster assembly proteins.

**Table 7 cancers-14-05768-t007:** Roles of altered DNA methylation in Cd-induced carcinogenesis.

Cells or Tissues	Compound	Route	Exposure Time	Modification (s)	Role	Mechanism
Wistar rats C57BL/6 mice [170]	CdCl_2_	i.p. injection	4 w	Caspase-8 promoter hypermethylation	Tumor suppressive	Cd led to reduced caspase-8 due to promoter hypermethylation, leading to decreased hepatic apoptosis and increased preneoplastic lesions.
TRL 1215 cell [171]	CdCl_2_	Culture medium	24 h/ 1–10 w	Global DNA and ApoE promoter hypermethylation	Pro-oncogenic	Cd caused global DNA hypermethylation and hypermethylation of ApoE promoter, leading to lowered ApoE expression. Lowered TET1 mediated ApoE promoter hypermethylation.
TRL 1215 [173,174]	CdCl_2_	Culture medium	10 w			
MCF-10A cell [176]	CdCl_2_	Culture medium	40 w	Global DNA hypomethylation	Unknown	Cd induced global DNA hypomethylation and c-Myc and K-Ras overexpression, but neither their correlation nor the role of global DNA hypomethylation in Cd-induced breast cancer was studied.
16HBE [181]	CdCl_2_	Culture medium	Chronic exposure for 35 passages	Global and hMSH2, ERCC1, XRCC1, and hOGG1 promoter hypermethylation	Pro-oncogenic	Cd induced global DNA hypermethylation and hypermethylated promoters of DNA repair genes (hMSH2, ERCC1, XRCC1, and hOGG1), which caused reduced expression of these genes and thereby accumulation of DNA damage.
RWPE-1 cell [183]	CdCl_2_	Culture medium	10 w	Global and RASSF1A and p16 promoter hypermethylation	Pro-oncogenic	Cd induced global DNA hypermethylation and hypermethylation of the promoter of RASSF1A and p16, which led to markedly reduced expression of these two tumor suppressors.
Multiple melanoma cell [184]	CdCl_2_	Culture medium	48–72 h	p16INK4A and caspase-8 hypermethylation	Pro-oncogenic	Cd induced silencing of p16INK4A and caspase-8, attributed to promoter hypermethylation caused by increased activity of DNMTs.
HMy2.CIR lymphoblast cell [187]	CdCl_2_	Culture medium	48 h/ 3 months	p16 promoter hypermethylation	Pro-oncogenic	Cd caused the down-regulation of p16 via hypermethylation of the CpG island in its promoter.
RWPE-1 cell [189]	CdCl_2_	Culture medium	8 w	Hypomethylation of HYAL1 and S100P; hypermethylation of NTM and NES	Pro-oncogenic	Expression of HYAL1 and S100P was up-regulated likely due to promoter hypomethylation; expression of NTM and NES was decreased likely due to promoter hypermethylation. Aberrant expression of these genes may contribute to Cd carcinogenesis.
HepG2 MCF7 cells [192]	CdCl_2_	Culture medium	24–48 h	Global DNA hypomethylation Hypomethylation of PRMT5 and EZH2 promoters	Pro-oncogenic	Cd induced global DNA hypomethylation and hypomethylation of PRMT5 and EZH2 promoters to induce their transcription. Up-regulated PRMT5 and EZH2 in turn led to the increased global level of H4R3me2 and H3K27me3 (both repressive histone marks), which potentially silenced tumor suppressors through remodeling the chromatin.

Cd: cadmium; i.p. injection: intraperitoneal injection; ApoE: apolipoprotein E; TET1: tet methylcytosine dioxygenase 1; hMSH2: mutS homolog 2; ERCC1: ERCC excision repair 1; XRCC1: X-ray repair cross-complementing 1; hOGG1: 8-oxoguanine DNA glycosylase; RASSF1A: Ras association domain family member 1; DNMT: DNA methyltransferase; HYAL1: hyaluronidase 1; S100P: S100 calcium-binding protein P; NTM: neurotrimin; NES: nestin; PRMT5: protein arginine methyltransferase 5; EZH2: enhancer of zeste 2 polycomb repressive complex 2 subunit.

**Table 8 cancers-14-05768-t008:** Roles of altered histone modifications in Cd-induced carcinogenesis.

Cells or Tissues	Compound	Route	Exposure Time	Modification (s)	Role	Mechanism
BEAS-2B cell [193]	CdCl_2_	Culture medium	20 passages	Global alterations of H3K4me2 H3K36me3 H3K9acS10ph H4K5ac H4K8ac H4K12ac	Pro- oncogenic	Cd-transformed cells showed markedly decreased H3K4me2 and H3K36me3 and up-regulated H3K9acS10ph, H4K5ac, H4K8ac, and H4K12ac histone modifications. Inhibition of histone acetyltransferase activity led to suppressed cancer phenotypes of transformed cells, suggesting that histone hyperacetylation is involved in Cd carcinogenesis.
MCF-7 T47-D cells [194]	CdCl_2_	Culture medium	24–72 h	Decreased H3K27ac in ATG5 promoter	Pro- oncogenic	Cd caused decreased ACSS2 expression, resulting in lowered ATG5 expression by reducing the level of H3K27ac in the ATG5 promoter. The resultant inhibition of ATG5-dependent autophagic flux contributed to Cd carcinogenesis.
UROtsa cell [196]	CdCl_2_	Culture medium	N/A	Increased H4-ac and methylation of H3K4, H3K9, and H3K27	Pro- oncogenic	Cd or As caused increased H4 acetylation and H3K4 methylation (transcriptional activation marks) and increased H3K9 and H3K27 methylation (transcriptional repression marks) in the promoter of MT-3. This “bivalent” histone modifications status of MT-3 facilitated MTF-1 binding to MT-3 promoter to activate its transcription. Up-regulated MT-3 in turn contributed to Cd carcinogenesis.
BEAS-2B [197]	CdCl_2_	Culture medium	6–48 h/20 w	Increased global H3K4me3 and H3K9me2	Unknown	Cd exposure led to increased global H3K4me3 and H3K9me2 at 24 h and 4 w by inhibiting the activities of H3K4 and H3K9 demethylases, respectively. However, global H3K4me3 and H3K9me2 had no significant change from 8 to 20 weeks, suggesting that increased global H3K4me3 and H3K9me2 are involved in early events of Cd carcinogenesis. However, the exact role of this change was not studied.

Cd: cadmium; ACSS2: acyl-CoA synthetase short-chain family member 2; ATG5: autophagy-related 5; MT-3: metallothionein 3; MTF-1: metal-responsive transformation factor-1.

**Table 9 cancers-14-05768-t009:** Roles of altered miRNA in Cd-induced carcinogenesis.

Cells or Tissues	Compound	Route	Exposure Time	miRNA	Alteration	Role	Target (s)	Mechanism
IEC-6 cell [198]	CdCl_2_	Culture medium	6–24 h	miR-124-3p miR-370-3p	Up	Tumor suppressor	Bcl-2	Cd induced miR-124-3p and miR-370-3p, which promoted Cd-induced apoptosis by targeting Bcl-2.
PC12 cell [199]	CdCl_2_	Culture medium	24 h	miR-34a-5p	Up	Tumor suppressor	Sirt1	Cd-induced miR-34a-5p expression and up-regulated miR-34a-5p contributed to Cd-induced apoptosis and ferroptosis by targeting Sirt1.
BEAS-2B BEP2D cells [201]	CdCl_2_	Culture medium	1–24 h/ 72 h	miR-30e	Down	Tumor suppressor	SNAIL1	Down-regulated miR-30e led to increased SNAIL1 expression, which contributed to Cd-induced cancer through the induction of EMT.
HepG2 [202]	CdCl_2_	Culture medium	24 h	miR-34a miR-200a	Down	Tumor suppressor	SNAIL1	miR-34a and miR-200a were down-regulated, causing increased SNAIL1, which contributed to Cd-induced cancer by inducing EMT.
Balb/c3T3 cell [207]	Mixture of NaAsO_2_, CdCl_2_, and Pb(C_2_H_3_O_2_)_2_	Culture medium	16 d	miR-222	Up	Oncogenic	Rad51c	Elevated miR-222 down-regulated Rad51c expression and impaired DNA homologous recombination during the initiation stage of cell transformation.
HUVEC cell [216]	CdCl_2_	Culture medium	12–36 h	miR-101	Down	Anti-angiogenic	COX-2	Suppressed miR-101 induced COX2 expression and ER stress; up-regulated COX2 and ER stress led to increased VEGF protein levels, resulting in abnormal angiogenesis.
PC3 DU145 cells [217]	CdCl_2_	Culture medium	20 w	miR-128-3p	Down	Tumor suppressor	SLC7A11	Cd induced LncRNA OIP5-AS1, which served as an endogenous sponge of miR-128-3p to up-regulate the expression of SLC7A11, thereby inhibiting ferroptosis.
BEAS-2B [218]	CdCl_2_	Culture medium	12–20 w	miR-224-5p	Up	Oncogenic	QKI	Circular RNA SHPRH was down-regulated, leading to up-regulated miR-224-5p and resultant down-regulation of QKI to promote proliferation, EMT, migration and invasion, and anchorage-independent growth of cells.
T-47D [219]	CdCl_2_	Culture medium	72 h	miR-374c-5p	Down	Tumor suppressor	GRM3	Cd exposure caused reduced m6A modification of pri-miRNA-374c, resulting in miR-374c-5p down-regulation. Suppression of miR-374c-5p in turn promotes proliferation, migration, and invasion of cells by activating GRM3.

Cd: Cadmium; Sirt1: sirtuin 1; SNAIL1: snail family transcriptional repressor 1; EMT: epithelial-to-mesenchymal transition; Rad51c: RAD51 paralog C; HUVEC: Human umbilical vein endothelial cells; COX-2: cyclooxygenase-2; ER: endoplasmic reticulum; VEGF: vascular endothelial growth factor; SLC7A11: solute carrier family 7 member 11; QKI: QKI, KH domain containing RNA binding; GRM3: glutamate metabotropic receptor 3. HUVEC: human umbilical vein endothelial cells.

## Data Availability

The data presented in this study are openly available in Medline and Embase.

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
