# Peer review of "Epigenetic Regulation in Chromium-, Nickel- and Cadmium-Induced Carcinogenesis"

_cancers, 2022, doi:10.3390/cancers14235768_

Round 1

Reviewer 1 Report

In this manuscript, entitled “Epigenetic regulation and non-coding RNAs in metal-induced carcinogenesis and angiogenesis”, the authors described how epigenetic modifications derived from heavy metal exposure contribute to the development of cancer. Overall, this work is very informative as it represents an in-depth report on the most recent advancements in the understanding of the epigenetic modifications induced by chromium, nickel, and cadmium. However, some issues could be addressed in order to improve the understanding of the manuscript.

The major issues include:

1) The introduction of pictures in paragraphs “2.1. DNA methylation and demethylation”, “2.2 Histone modification”, and “2.3.1. miRNA” to better explain the illustrated concepts. For example, regarding paragraph “2.1. DNA methylation and demethylation”, a picture showing how the DNA methylation takes place might help the reader to better understand the mechanism. Given that in this paragraph are introduced the mechanisms that will be discussed in the core of the paper, I feel that these changes might help improve the overall understanding of the article. Moreover, this might also allow non-experts in the field to grasp the basic concepts and therefore to benefit from this review.

2) I would recommend inserting some tables at the end of each paragraph treating the epigenetic effects of the metals. These tables should summarize the information provided and could be made similarly to those already present in the manuscript (namely tables 1, 2, and 3). This is highly needed as it helps the reader to extract from the article the most relevant information for his research.

3) Regarding the tables, to improve the readability of the column “mechanism” it is recommended to add more space or to draw a horizontal line between the different miRNAs.

Since the work is well-conceived and clearly discussed, I believe that the requested implementation will be beneficial for the overall importance of this manuscript, and thus, worthy of publication in Cancers.

Finally, some other minor issues include:

1) The introduction of more basic information in the introduction of each of the heavy metals (namely paragraphs “3. Chromium (Cr)”, “4. Nickel (Ni)”, and “5. Cadmium (Cd)”). In particular, it could be reported where these heavy metals tend to accumulate in the body and some basic info on the clinical protocols (if present) to treat poisoning and tumours induced by these metals.

2) In paragraph “4. Nickel (Ni)” it could be indicated what are the main sources of exposure to this metal as well as what are the signs of intoxication (as they were discussed in paragraph “3. Chromium (Cr)”). Similarly, in paragraph “5. Cadmium (Cd)” a few more lines could be added to describe more in depth the signs of intoxication.

3) Although this review is widely exhaustive with regard to the treated metals, since it claims to tackle environmental and occupational exposure, it could also provide some information on other well-known pollutants, such as lead and mercury.

4) In line 167 it says “miRNAs play key roles in cancer development, angiogenesis, and drug resistance”. In the next lines an example of drug resistance caused by miRNAs could be provided.

Author Response

1) The introduction of pictures in paragraphs “2.1. DNA methylation and demethylation”, “2.2 Histone modification”, and “2.3.1. miRNA” to better explain the illustrated concepts. For example, regarding paragraph “2.1. DNA methylation and demethylation”, a picture showing how the DNA methylation takes place might help the reader to better understand the mechanism. Given that in this paragraph are introduced the mechanisms that will be discussed in the core of the paper, I feel that these changes might help improve the overall understanding of the article. Moreover, this might also allow non-experts in the field to grasp the basic concepts and therefore to benefit from this review.

Answer: Thank you for the valuable suggestions. We drafted the illustrations of DNA methylation and demethylation, Histone modification, and miRNA to help readers understand these concepts. The illustrating figures are inserted below the corresponding paragraphs.

2) I would recommend inserting some tables at the end of each paragraph treating the epigenetic effects of the metals. These tables should summarize the information provided and could be made similarly to those already present in the manuscript (namely tables 1, 2, and 3). This is highly needed as it helps the reader to extract from the article the most relevant information for his research.

Answer: We added tables to summarize DNA methylation and histone modification for each metal carcinogen based on the reviewer’s suggestion .

3) Regarding the tables, to improve the readability of the column “mechanism” it is recommended to add more space or to draw a horizontal line between the different miRNAs.

Answer: We made the “mechanism” column wider for more space, and added a horizontal line between different miRNAs.

Finally, some other minor issues include:

1) The introduction of more basic information in the introduction of each of the heavy metals (namely paragraphs “3. Chromium (Cr)”, “4. Nickel (Ni)”, and “5. Cadmium (Cd)”). In particular, it could be reported where these heavy metals tend to accumulate in the body and some basic info on the clinical protocols (if present) to treat poisoning and tumours induced by these metals.

Answer: Thank you for the suggestions. We have added more information regarding body accumulation and signs of intoxication in the introduction for each metal.

2) In paragraph “4. Nickel (Ni)” it could be indicated what are the main sources of exposure to this metal as well as what are the signs of intoxication (as they were discussed in paragraph “3. Chromium (Cr)”). Similarly, in paragraph “5. Cadmium (Cd)” a few more lines could be added to describe more in depth the signs of intoxication.

Answer: We have added more information about Ni exposure, Ni intoxication, and Ni-induced human diseases based on the reviewer’s comments.

3) Although this review is widely exhaustive with regard to the treated metals, since it claims to tackle environmental and occupational exposure, it could also provide some information on other well-known pollutants, such as lead and mercury.

Answer: We thank the reviewer’s suggestion that other well-known pollutants such as lead and mercury, should be comprehensively discussed. However, due to the page limit of this article, we will summarize the most up-to-date studies on these pollutants in a separate review article in the future.

4) In line 167 it says “miRNAs play key roles in cancer development, angiogenesis, and drug resistance”. In the next lines an example of drug resistance caused by miRNAs could be provided.

Answer: We added an example of miRNA-induced drug resistance in the revised manuscript.

Reviewer 2 Report

The article by Lei Zhao is a well written and comprehensive review, contributing to a broader readership to the journal. The authors' team is well established in the field; therefore, I highly recommend the acceptance of the manuscript in its present form.

Author Response

We thank reviewers 2 for the strong positive opinions on the manuscript

Reviewer 3 Report

The manuscript provides a comprehensive summary of the most recent literature on epigenetic changes induced by heavy metals including chromium, nickel, and cadmium. They first briefly introduced several epigenetic mechanisms such as DNA methylation, histone modifications, and non-coding RNAs focusing on miRNA, lncRNA, and cirRNA and then highlighted the recent findings of the changes in these modifications responding to Cr, Ni, and Cd exposures and their implications in carcinogenesis. The authors also touch upon study limitations and future perspectives. In total, this is a well-organized timely review that will be of interest to many researchers.

Specific comments:

1.     Title: expression of non-coding RNAs is also belong to a epigenetic regulation and this review is mostly about carcinogenesis, thus the title could be changed to something like “Epigenetic regulation in chromium, nickel and cadmium-induced carcinogenesis”.

2.     All tables: it would be useful and important to add compound types, exposure time, route, and tissue types (cell, mice, or human etc) to all the tables.

3.     Page 7, lines 292-310: describes histone modification changes and chromatin structural changes respectively under the subtitle of “DNA methylation”, therefore the first paragraph should be moved to following section and the second paragraph should be removed.

4.     cirRNA change is only described in the Cd part. That is the case for RNA m6A modification as well. And RNA m6A modification was not mentioned in “2. Epigenetics” section. 

5.     “2. Epigenetics” should be changed to “2. Epigenetic mechanisms”.

6.     Each “Histone modification” section is too long. To make it more readable, this review suggests to divide each “Histone modification” section to two parts, i.e., “histone acetylation” and “histone methylation”.

Author Response

1)  Title: expression of non-coding RNAs is also belong to a epigenetic regulation and this review is mostly about carcinogenesis, thus the title could be changed to something like “Epigenetic regulation in chromium, nickel and cadmium-induced carcinogenesis”.

Answer: We changed the title to “Epigenetic regulation in chromium-, nickel- and cadmium-induced carcinogenesis” based on the reviewer’s suggestion.

2)  All tables: it would be useful and important to add compound types, exposure time, route, and tissue types (cell, mice, or human etc) to all the tables.

Answer: We have revised the table by adding available information about compound types, exposure time, route, and tissue types (cell, mice, or human etc) to all the tables.

3) Page 7, lines 292-310: describes histone modification changes and chromatin structural changes respectively under the subtitle of “DNA methylation”, therefore the first paragraph should be moved to following section and the second paragraph should be removed.

Answer: Lines 296-303 discussed histone modification of MLH1 in Cr(VI)-exposed cells. This paragraph has been moved to the histone modification section; Lines 304-313 described the effect of chromatin structural changes on gene expression. This paragraph is neither DNA methylation nor histone modification, and we deleted it.

4)  cirRNA change is only described in the Cd part. That is the case for RNA m6A modification as well. And RNA m6A modification was not mentioned in “2. Epigenetics” section.

Answer: Because cirRNA is only applied to Cd rather than to all metals, I would like to move the introduction of cirRNA from the general introduction section (1.3.3 circRNA) to the Cd section (4.3.3). I have briefly introduced cirRNA before discussing its role in Cd-induced carcinogenesis. The introduction of m6A modification will stay in the Cd section, paragraph 4.3.4.

5)  “2. Epigenetics” should be changed to “2. Epigenetic mechanisms”.

Answer: Thank you for the suggestion. We have changed it to “Epigenetic mechanisms”.

6)   Each “Histone modification” section is too long. To make it more readable, this review suggests to divide each “Histone modification” section to two parts, i.e., “histone acetylation” and “histone methylation”.

Answer: We have split the “Histone modification” for Cr(VI) and Ni exposure into “histone acetylation” and “histone methylation” parts. But the histone modifications for Cd is a short paragraph, and the studies investigated both methylation and acetylation together, which can’t be clearly separated. So we didn’t split the Histone modification paragraph for Cd.

Reviewer 4 Report

Chronic environmental exposure to heavy metals significantly contributes to the development of human malignancies. It is estimated that a significant fraction of cancer deaths results from an uncontrolled angiogenic process, which can be initiated by deregulated epigenetic process during cancer development and progression. Many studies have shown that heavy metals alter epigenetic regulation, leading to expression of non-coding RNAs responsible for regulating expression of genes involved in angiogenesis. Given the importance of epigenetic mechanisms in regulating cell proliferation and differentiation after exposure to toxic metals, the current review is timely and important. 

The review manuscript was well organized and clearly written, It covers all important aspects of carcinogenicity of environmental metal toxicants. Significantly, it focuses on roles of DNA modification, histone modifications, and non-coding RNAs in transformation and angiogenesis initiated by exposure to metals. It is a great resource for scientists in the metal field to get updated information about deregulation of angiogenesis in metal carcinogenesis. 

Author Response

We thank reviewers 4 for the strong positive opinions on the manuscript

Round 2

Reviewer 1 Report

All major and minor issues raised have been addressed satisfactorily. Therefore, I recommend the publication of this manuscript in its current form.